# OVO positively regulates essential maternal pathways by binding near the transcriptional start sites in the *Drosophila* female germline

**Leif Benner[1,2]\*, Savannah Muron[1], Jillian G Gomez[1], Brian Oliver[1]**

[1]Section of Developmental Genomics, Laboratory of Biochemistry and Genetics, National Institute of Diabetes and Digestive and Kidney Diseases, National Institutes of Health, Bethesda, United States; [2]Department of Biology, Johns Hopkins University, Baltimore, United States

**\*For correspondence:**
leif.benner@gmail.com

**Competing interest:** The authors declare that no competing interests exist.

**Abstract** Differentiation of female germline stem cells into a mature oocyte includes the expression of RNAs and proteins that drive early embryonic development in *Drosophila*. We have little insight into what activates the expression of these maternal factors. One candidate is the zinc-finger protein OVO. OVO is required for female germline viability and has been shown to positively regulate its own expression, as well as a downstream target, *ovarian tumor*, by binding to the transcriptional start site (TSS). To find additional OVO targets in the female germline and further elucidate OVO's role in oocyte development, we performed ChIP-seq to determine genome-wide OVO occupancy, as well as RNA-seq comparing hypomorphic and wild type rescue *ovo* alleles. OVO preferentially binds in close proximity to target TSSs genome-wide, is associated with open chromatin, transcriptionally active histone marks, and OVO-dependent expression. Motif enrichment analysis on OVO ChIP peaks identified a 5'-TAACNGT-3' OVO DNA binding motif spatially enriched near TSSs. However, the OVO DNA binding motif does not exhibit precise motif spacing relative to the TSS characteristic of RNA polymerase II complex binding core promoter elements. Integrated genomics analysis showed that 525 genes that are bound and increase in expression downstream of OVO are known to be essential maternally expressed genes. These include genes involved in anterior/posterior/germ plasm specification (*bcd, exu, swa, osk, nos, aub, pgc, gcl*), egg activation (*png, plu, gnu, wisp, C(3)g, mtrm*), translational regulation (*cup, orb, bru1, me31B*), and vitelline membrane formation (*fs(1)N, fs(1)M3, clos*). This suggests that OVO is a master transcriptional regulator of oocyte development and is responsible for the expression of structural components of the egg as well as maternally provided RNAs that are required for early embryonic development.

## eLife assessment

This **useful** manuscript extends prior work to identify OVO as a major transcriptional activator of the female germline gene expression program. Using a combination of **solid** genomic strategies, the authors demonstrate that OVO binds to the promoters of hundreds of genes in the female germline and promotes their expression.

## Introduction

*Drosophila* early embryonic development is directed by events that take place during oogenesis. Germline stem cells (GSCs) asymmetrically divide to renew the stem cell population and send one

daughter cell towards oogenesis. In the germarium, oogenesis is preceded by four rounds of incomplete mitotic divisions resulting in a 16-cell egg chamber. One cell is specified as the oocyte, which is arrested in prophase of meiosis I, while the rest of the 15 cells enter endoreplication cycles and become nurse cells (NCs). Once the 16-cell egg chamber buds from the germarium, the NCs begin to transcribe and translate a vast array of RNAs and proteins that serve diverse functional roles (*Bastock and St Johnston, 2008*; *Spradling et al., 2022*). These roles include positioning maternal mRNAs and proteins in the correct spatial orientation to support anterior/posterior, and dorsal ventral axis specification, as well negative regulators of translation to ensure that the maternal mRNAs are not translated before fertilization (*Lasko, 2012*). The oocyte also contains a number of proteins and mRNAs that are needed for egg activation, completion of meiosis and initiation of embryonic development after fertilization (*Avilés-Pagán and Orr-Weaver, 2018*). To repeat the process of oogenesis from generation to generation, germ cells in the developing embryo need to be specified and maintained separately from the rest of the developing somatic cell population. This requires maternally controlled localization of the germ plasm and early pole cell formation in the embryo (*Mahowald, 2001*). While the complex interactions between maternally supplied mRNAs and proteins have been well studied, transcriptional regulation driving the expression of these pathways are less well understood.

Few positive regulators of female-specific germ cell transcription have been identified. Genes such as *grauzone* (*grau*) and *maternal gene required for meiosis* (*mamo*) have been shown to activate the transcription of *cortex* and *vasa*, respectively, in the female germline (*Harms et al., 2000*; *Nakamura et al., 2019*). Active repression of male-specific transcription through the activity of *egg*, *wde*, and *Su(var)205* (*Smolko et al., 2018*), or global repression of non-ovarian transcriptional networks through the function of *sov* (*Benner et al., 2019*), have shown the importance of heterochromatin formation in the female germline for cellular identity and oocyte development. In fact, recent work has shown the importance of transcriptional repression, mediated through changes in histone modifications, as a key regulator of egg chamber differentiation. GSCs have been shown to exist in a sort of 'ground state' of histone modifications. Characterized with modest non-canonical repressive H3K9me3 and H3K27me3 histone marks at many genes, as well as transcriptionally active H3K27ac histone marks and open chromatin at others (*Pang et al., 2023*; *DeLuca et al., 2020*). As oocyte development continues, repressive histone marks associated with heterochromatin begin to increase in abundance resulting in fewer histone marks associated with transcription and open chromatin. This suggests that gene expression becomes more restricted throughout the differentiation process. However, it is unlikely that the female germline directs oocyte development solely through a repressive transcriptional model. Whether the female germline expresses paralogs of the RNA polymerase II complex, like the male germline (*Hiller et al., 2004*; *Hiller et al., 2001*; *Lu et al., 2020*), or if there are pioneering transcription factors involved in determining the open chromatin status for female germ cell-specific expression, or something else entirely, has yet to be determined.

Although few female-specific germline transcription factors have been identified, the conserved zinc-finger transcription factor OVO has long been known to be required for female germ cell viability. Female germ cells that are *ovo⁻* do not survive into adulthood (*Oliver et al., 1987*; *Oliver et al., 1990*; *Benner et al., 2023*). Hypomorphic *ovo* alleles, specifically ones that disrupt the transcriptional activator OVO-B, show an arrested egg chamber phenotype, indicating that wild-type OVO-B activity is required for oocyte maturation (*Salles et al., 2002*; *Benner et al., 2023*). Germline OVO is expressed at all stages of oogenesis, where it is eventually maternally loaded into the egg. Maternal OVO becomes specifically localized to the developing germline and persists throughout embryogenesis until zygotic OVO is expressed (*Hayashi et al., 2017*; *Benner et al., 2023*). Thus, OVO is eternally expressed in the female germline, suggesting it may be a key regulator of female-specific germline transcription. However, only two downstream targets have previously been identified for OVO. OVO has been shown to positively regulate the expression of its own transcription, therefore executing an autoregulatory loop, as well as positively regulating the transcription of the gene *ovarian tumor* (*otu*)(*Lü et al., 1998*; *Bielinska et al., 2005*; *Lü and Oliver, 2001*; *Andrews et al., 2000*). *otu* is also required in the female germline, where *otu⁻* germ cells show viability and germline tumor phenotypes (*Bishop and King, 1984*). The *ovo* phenotype is epistatic to that of *otu*, and ectopic *otu* expression cannot rescue female germ cell death due to loss of *ovo* (*Hinson et al., 1999*; *Pauli et al., 1993*). Therefore, *ovo* must be responsible for activating the transcription of genes in addition to *otu* for female germ cell survival and differentiation.

We expanded our knowledge of OVO's role in the female germline by determining genome-wide OVO occupancy and global transcriptional changes downstream of OVO. This allowed us to determine which genes OVO binds, and which genes transcriptionally respond to OVO in vivo. We show that OVO is directly regulating essential maternal pathways such as axis specification, primordial germ cell formation, egg activation, and maternal mRNA translation regulation. Together, we show that OVO plays a pivotal role in the positive transcriptional regulation of oocyte and early embryonic development. We show that OVO likely carries out this regulation by binding at or in close proximity to the promoters of the genes it regulates and that OVO DNA binding motifs are enriched at or near the transcriptional start site (TSS) of OVO-responsive genes, although the spacing of OVO-binding sites suggests that it is not a component of the RNA polymerase complex. OVO binding is also a signature of open chromatin status and active transcription throughout oocyte differentiation. Altogether, we suggest that OVO is required for the activity of a large number of female germline promoters and is likely a key regulator of oocyte maturation and RNAs and proteins that are required for early embryonic development.

## Results

### OVO binds promoters genome-wide

OVO-B, the predominant protein isoform expressed from the *ovo* locus in the female germline (*Benner et al., 2023*), is a positive regulator of transcription (*Andrews et al., 2000*) at both the *otu* and *ovo* locus. Transgenic reporter constructs of *otu* and *ovo* require OVO-binding sites both at and upstream of the TSS in order to recapitulate full reporter expression (*Bielinska et al., 2005*; *Lü et al., 1998*; *Lü and Oliver, 2001*). Females hemizygous for antimorphic dominant gain-of-function (*ovo^D*) or homozygous recessive (*ovo^D1rv*) *ovo* alleles lack germ cells in the adult ovary (*Oliver et al., 1987*; *Benner et al., 2023*). True OVO-B null alleles created by deletion of the *ovo-B* promoter have the same germ cell loss phenotype (*Benner et al., 2023*). The phenotypes of *otu⁻* females range from germ cell death to ovarian tumors depending on the allele and undefined stochastic factors (*Bishop and King, 1984*). It is possible that the germ cell death phenotype in *ovo⁻* female germ cells can solely be explained by failure of OVO to activate *otu* expression, however, this is highly unlikely. The *ovo^D1rv* phenotype is epistatic to *otu⁻*, and ectopic *otu⁺* in *ovo⁻* germ cells does not rescue the germ cell death phenotype (*Hinson et al., 1999*). This suggests that OVO regulates the expression of additional genes in the female germline.

We wanted to identify OVO target genes in the female germline by using two complementary genome-wide approaches to test for OVO presence and function. Specifically, we determined OVO occupancy genome-wide with ChIP-seq, and determined *ovo* function by comparing the RNA expression profiles between *ovo⁺* and *ovo* hypomorphs in the female germline. In order to determine OVO occupancy genome-wide, we performed ChIP-seq on adult ovaries in triplicate, using two C-terminally tagged alleles as affinity purification tools (*ovo^Cterm-3xFHA* i.e. OVO-HA and *ovo^Cterm-GFP* i.e. OVO-GFP, *Figure 1A–C*, *Figure 1—figure supplement 1A*; *Benner et al., 2023*) and called significantly enriched peaks from OVO-HA and OVO-GFP compared to their respective input controls.

We first compared the pulldown results with the OVO-HA versus OVO-GFP ChIP reagents. The GFP pull down appeared to be more efficient, but nevertheless we found excellent agreement, as most OVO-HA peaks were also found in the OVO-GFP dataset. The OVO-GFP ChIP dataset had 7235 significant ChIP peaks according to peak enrichment analysis genome-wide, while OVO-HA ChIP dataset had 3393 significant peaks genome-wide (*Supplementary file 1*). To determine the similarity in significant peak calling between the two datasets, we calculated a Jaccard index (intersection/union) between the significantly enriched peaks from the tagged *ovo* allele bearing ovaries. The Jaccard index between OVO-HA and OVO-GFP ChIP peaks was 0.64 (where 0=no overlap and 1=full overlap) with a total of 3094 ChIP peaks overlapping. Thus, almost all significant OVO-HA ChIP peaks were also found within the OVO-GFP ChIP dataset (91% of OVO-HA peaks overlapped OVO-GFP peaks). OVO-GFP pulldown was either more effective, or less likely, promiscuous. Regardless, we decided to use the conservative intersection of the two datasets (3094 peaks) for downstream OVO occupancy informatics (*Supplementary file 2*).

OVO is a sequence-specific DNA-binding protein, but many transcription factors also have a more general affinity for DNA. Additionally, immunoprecipitation can capture indirect interactions due to

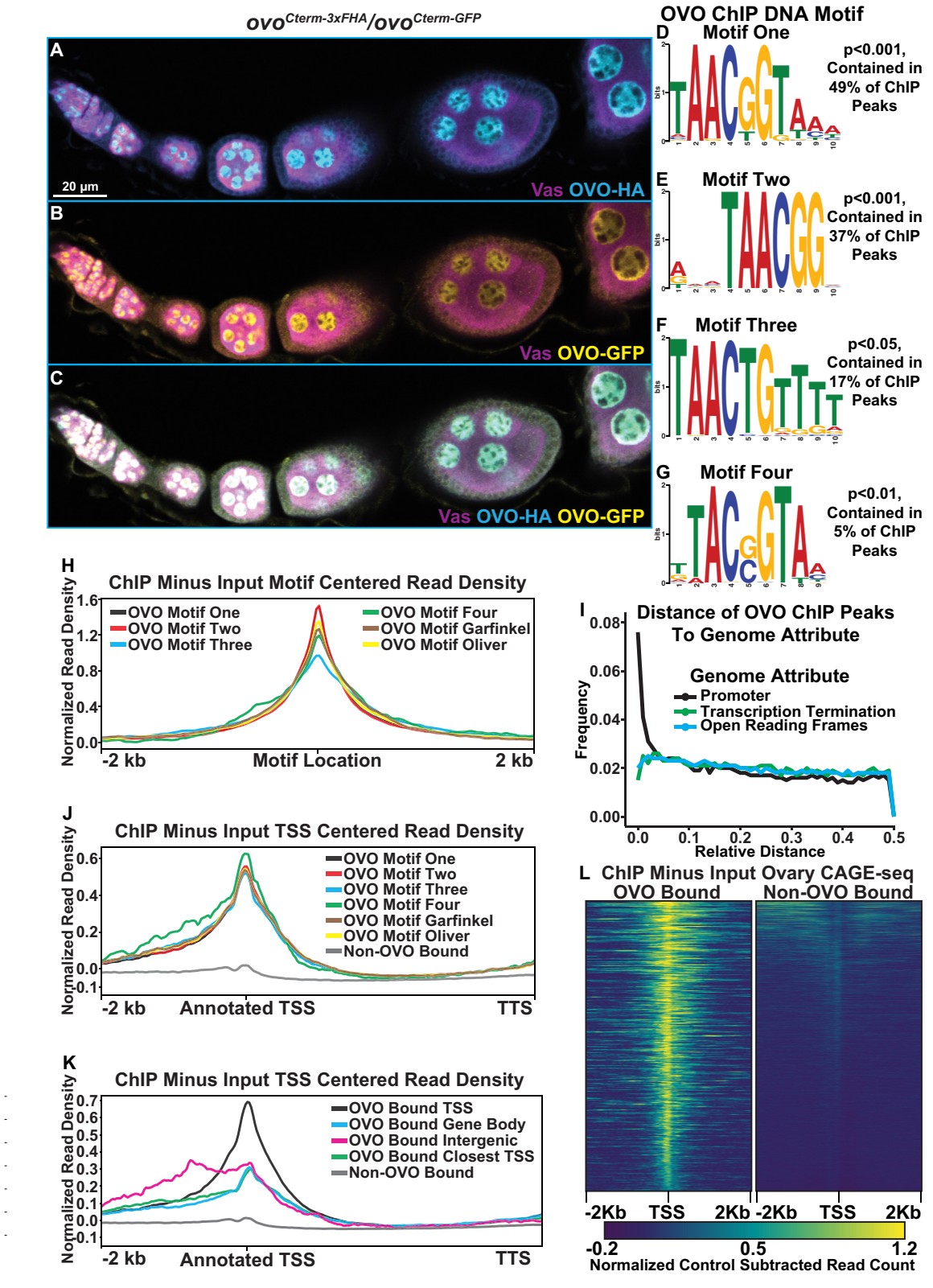

**Figure 1.** Significantly enriched OVO DNA binding motifs and OVO ChIP attributes genome-wide. (**A–C**) Immunofluorescent staining of adult ovarioles of *ovo*$^{Cterm-3xFHA}$/*ovo*$^{Cterm-GFP}$ females (20 x, scale bar = 20 µm). Ovarioles were stained for Vas (purple) to label the germline, HA (cyan) to label OVO-HA, and GFP (yellow) to label OVO-GFP. The homozygous version of these alleles were used to ChIP OVO. (**D–G**) Significantly enriched motifs found within overlapping OVO ChIP peaks. The percentage of OVO ChIP peaks containing each motif and their corresponding p-value are indicated to the

*Figure 1 continued on next page*

*Figure 1 continued*

right. (**H**) OVO ChIP minus input control ChIP-seq read coverage density centered on the location of the four de novo OVO DNA binding motifs and previously defined in vitro OVO DNA binding motifs (*Lü et al., 1998*; *Bielinska et al., 2005*; *Lee and Garfinkel, 2000*). (**I**) Relative distance of OVO ChIP peaks to gene level promoters, terminations sequences, and open reading frames genome-wide. (**J**) OVO ChIP minus input control ChIP-seq read coverage density for genes containing significant OVO ChIP peaks and the corresponding OVO DNA binding motif. Genes are centered on the transcriptional start site. (**K**) OVO ChIP minus input control ChIP-seq read coverage density for genes bound by OVO over the TSS, gene body, closest TSS in intergenic space, closest TSS for all, or not bound. Genes are centered on the transcriptional start site. (**L**) OVO ChIP minus input control ChIP-seq heatmaps centered on the dominant significant ovary CAGE-seq TSSs overlapping or not overlapping OVO ChIP peaks.

The online version of this article includes the following figure supplement(s) for figure 1:

**Figure supplement 1.** Significant OVO DNA binding motif, enriched motifs in OVO ChIP peaks, and ATAC and H3K27ac ChIP-seq read density for OVO bound and not bound genes.

nuclear topology in addition to direct sequence-specific binding. For example, in the particular case of OVO TSS binding, this could be direct, as shown in the case of *ovo* and *otu* loci (*Lü et al., 1998*; *Lü and Oliver, 2001*; *Bielinska et al., 2005*), or could be due to looping of an OVO-bound enhancer to the core promoter. Determining if there are canonical OVO-binding sites at peaks can help to distinguish direct and indirect binding. If OVO is directly binding to the TSS, we would expect to find OVO binding site enrichment at that location. To examine the sequences enriched in peaks, we looked directly for the known OVO-binding sites previously defined by footprinting (*Lü et al., 1998*) and SELEX-seq (*Bielinska et al., 2005*; *Lee and Garfinkel, 2000*). We also did de novo motif finding on this substantial dataset to refine the sequence-specific motifs bound by OVO and perhaps other motifs associated with other transcription factors preferentially bound by OVO enhancers, or OVO proximal sites in 3D nuclear space. We performed novel motif enrichment analysis using STREME (*Bailey, 2021*) with our overlapping ChIP peaks and found a number of significant motifs within our dataset. The most significant motif, 5'-TAACGGTAAA-3', was found in 49% of significant OVO ChIP peaks (*Figure 1D*, 'Motif One'). This motif is highly similar to the OVO DNA binding motif that has been reported twice before, 5'-AGTAACNGT-3' (SELEX method, 'Garfinkel OVO Motif') and 5'-TGTA ACNGT-3' (Footprinting method, 'Oliver OVO Motif'). The only differences between motif one in our dataset and the literature, is that the de novo motif is two nucleotides shorter than the previously described motifs at the 5' end, extends 3 nucleotides downstream, and includes a second G near the 3'-end (*Figure 1—figure supplement 1B*). Collectively, this is strong evidence that the core OVO-binding sequence is 5'-TAACNGT-3'. Some binding sites can be recognized by multiple transcription factors. To determine if other characterized transcription factors recognize this sequence, we searched for significant matches to motif one in the Jaspar database (*Castro-Mondragon et al., 2022*) of known *Drosophila* motifs using Tomtom (*Gupta et al., 2007*). The OVO DNA binding motif ('Garfinkel OVO Motif') was scored as a significant match (*Figure 1—figure supplement 1B*, $P<0.05$). Along with the OVO DNA binding motif, other motifs were also significantly enriched in OVO ChIP peaks. The motif 5'-GWGMGAGMGAGABRG-3' (*Figure 1—figure supplement 1C*) was found in 18% of OVO ChIP peaks and is a significant match to the DNA binding motifs of the transcription factors GAF (*Trl*) (*Omelina et al., 2011*) and CLAMP (*Soruco et al., 2013*). *Trl* germline clones are not viable, indicating that GAF activity is required in the germline during oogenesis (*Chen et al., 2009*). The possibility that OVO binds with and regulates genes alongside of GAF given the enrichment of both transcription factors DNA binding motifs is intriguing. Other significantly enriched motifs 5'-ACACACACACACACA-3' (29% of peaks, *Figure 1—figure supplement 1D*), 5'-RCAACAACAACA ACA-3' (26% of peaks, *Figure 1—figure supplement 1E*), and 5'-GAAGAAGAAGAAGAR-3' (17% of peaks, *Figure 1—figure supplement 1F*) were present in OVO ChIP peaks, however, these motifs did not significantly match known DNA binding motifs of other transcription factors. Determining the factors that bind to these sequences will certainly help elucidate our understanding of transcriptional control with relationship to OVO in the female germline.

Not every peak region had one of the consensus OVO DNA binding motifs. This does not mean a priori that they bound OVO indirectly. Motif enrichment can be driven by a few strongly enriched sequences, so that more minor enrichments of variants are missed. Therefore, we carefully examined the 51% of our overlapping OVO ChIP peaks where OVO DNA binding motif one was not found. This second round of de novo analysis revealed enrichment of OVO DNA binding motif derivatives. For example, the third most significant motif (found in 37% of peaks) was 5'-RWMTAACGGV-3'

(*Figure 1E*, motif two). This motif had the core 5'-TAACNGT-3' sequence found in all three aforementioned methods, however, the last nucleotide in the core sequence is unspecific and lacks the three 3' nucleotides found in motif one. Two other derivative motifs, 5'-TAACTGTTTT-3' (found in 17% of sequences, *Figure 1F*, motif three), and 5'-TTACSGTAA-3' (found in 5% of sequences, *Figure 1G*, motif four), vary within the central core motif (at positions 5 and 2 of the core sequence, respectively) and upstream and downstream ends. Searching for all variations of the OVO DNA binding motif (*Supplementary file 3*) within our significant overlapping ChIP peaks indicated that 72% of peaks contained at least one variation of these four binding motifs. It is a reasonable hypothesis that OVO peaks are most often due to direct, rather than indirect OVO binding.

A prediction for direct OVO binding to motifs, is that the motif should be centered within the peak of fragmented input DNA sequences. Therefore, we plotted the significant ChIP (minus input) read density centered on the location of the motif. We found that the read density for all ChIP peaks that contain each one of the de novo OVO motifs, as well as the in vitro OVO motifs, are centered over the motif location (*Figure 1H*). This suggests that all of these motifs from our analysis are bonafide OVO DNA binding sites in vivo. While it is possible that OVO comes into contact with regions of DNA in three-dimensional nuclear space non-specifically, the presence of OVO motifs within a large percentage of significant ChIP peaks in vivo and enrichment of OVO ChIP read density at the location of the motifs, strongly reinforces the idea that our OVO ChIP dataset contains regions centered on sequences specifically bound by OVO in the ovary.

Given the clear function of OVO occupancy near the TSS of its two known targets, *ovo* and *otu* (*Lü et al., 1998*; *Bielinska et al., 2005*; *Lü and Oliver, 2001*), we were interested in determining if OVO peaks are generally near the TSS of other target genes as well. In addition to informing the biology of OVO function, this simplifies the problem of associating peaks to potential functional target genes. As a preliminary test of this idea, we determined if the fully overlapping OVO-HA and OVO-GFP peaks were spatially enriched with respect to the currently annotated gene model elements such as TSS, openreading frames (ORFs), or transcription termination sequences (TTS). If the TSS association of OVO at the two known targets reflects a general propensity then we expect OVO ChIP peaks to be more closely associated with TSS than other gene elements. To carry out this analysis, we normalized the genome for these three gene elements, such that the distance between adjacent loci was 1. If there is no enrichment for OVO peaks to a specific gene element then the peak location would have an equal frequency from 0.0 to 0.5 relative distance. Measuring the relative distance of our OVO ChIP peaks to TSS, ORFs, and TTS, showed that the OVO binding was highly enriched near TSS/promoter locations and was not correlated with ORF and TTS locations (*Figure 1I*). These results confirmed that OVO is characterized by core promoter proximal binding. Since OVO ChIP peaks as a class are associated with TSS, we plotted the ChIP minus input read density of genes that overlap significant ChIP peaks to examine the full distribution. We found that the OVO ChIP read density was highly enriched over the TSS and was not due to a few highly enriched examples (*Figure 1J*). This builds on the idea that OVO is binding directly over, or in close proximity to the TSS of its target genes genome-wide. In other words, the previous work showing OVO binding the *ovo-B* and *otu* TSS (*Bielinska et al., 2005*; *Lü et al., 1998*; *Lü and Oliver, 2001*) is typical. This very specific binding of OVO to the TSS is intriguing and unusual, as this region associates with the basal transcriptional machinery. It raises the possibility that OVO is not a typical transcription factor that acts primarily via enhancer binding but might be part of the core promoter binding complex or acts to precondition the core promoter region for example.

Although OVO ChIP peaks overlapping genes showed a strong read density enrichment over the TSS, we found that only 45% (1394/3094) of OVO ChIP peaks directly overlapped a TSS. 43% (1339/3094) of OVO ChIP peaks were found to overlap the gene body downstream of the TSS (intronic and exonic sequences) and 12% (366/3094) did not overlap any gene elements, indicating that they were intergenic. We were interested in the differences between OVO binding directly over the TSS or at more distal upstream and downstream sites. We decided to plot the OVO ChIP read density of these different classes of OVO-binding patterns and found that OVO bound over the TSS produced a sharp read density enrichment over the TSS which was consistent with what was found for all OVO-bound genes (*Figure 1K*). OVO binding along the gene body surprisingly also showed a read density enrichment over the TSS, although the magnitude of read density enrichment was notably less than TSS OVO binding. Intergenic OVO binding also showed these same characteristics

with a notable upstream read density enrichment possibly indicative of enhancer binding. This indicates that although the significantly called OVO ChIP peaks did not overlap the TSS, there was still a propensity for TSS sequences to be enriched with OVO ChIP over the input control. This could be due to weaker direct in vivo binding of OVO to these TSSs or indirect interactions between the upstream/downstream OVO bound sequences and the TSS, possibly through a looping enhancer-promoter interaction. However, regardless of the location of the OVO ChIP peak, OVO seemed to always be enriched at or in close proximity to TSSs.

The OVO ChIP read density was highly enriched over the annotated TSS of target genes, but TSS annotation is challenging and can be tissue specific. We were interested in empirically determining if the same enrichment was present in TSSs utilized specifically in ovarian tissue. The 5' ends of mRNA are capped. In order to determine where these caps mapped to the genome, we analyzed Cap Analysis of Gene Expression (CAGE-seq) data from adult *Drosophila* ovaries (*Chen et al., 2014*) and extracted the dominant significant TSSs in the ovary. CAGEr predicted 6,856 significant TSSs in the ovary dataset, of which 1047 overlapped with OVO ChIP peaks. We plotted the OVO ChIP minus input read density centered on the significant ovary CAGE-seq TSSs for TSSs that overlapped or did not overlap OVO ChIP peaks (*Figure 1L*). We found that OVO ChIP read density was highly enriched over the location of TSSs from ovary CAGE-seq that overlapped OVO ChIP peaks when compared to TSSs that did not overlap OVO ChIP peaks. Thus, OVO TSS binding is not due to poor annotation of ovarian TTSs. Furthermore, OVO is binding at or near TSSs of genes actively being transcribed in the ovary.

## OVO binding is associated with open chromatin and transcriptionally active histone marks

Our OVO ChIP data indicated that OVO was binding at or in close proximity to promoters genome-wide. OVO could have a positive and/or negative effect on transcription at these locations. For example, OVO could help recruit or sterically hinder RNA polymerase binding to TSSs. However, previous *ovo* reporter constructs show positive effects of OVO binding near TSS (*Lü and Oliver, 2001*; *Lü et al., 1998*; *Bielinska et al., 2005*). If OVO binding is generally promoting transcription then we hypothesize that it would be more closely associated with histone marks associated with active transcription, such as H3K27ac and H3K4me3, as well as lower nucleosome density that can be measured through ATAC-seq. In contrast, OVO binding would be expected to negatively correlate with repressive H3K9me3 and H3K27me3 histone marks and higher nucleosome density. It is technically difficult to determine changes in chromatin status and transcription in germ cells that lack OVO, as the phenotype is cell death (although we will return to this later for transcription profiling) but analyzing OVO binding in the context of ovarian chromatin was highly informative.

Recent work profiling nucleosome density and histone marks have shown that female GSCs have a 'ground state' chromatin profile (*DeLuca et al., 2020*), similar to the histone mark profiles that are found in early embryos (*Li et al., 2014*). This has been characterized to contain non-canonical H3K27me3 profiles and low H3K9me3 histone levels (*Pang et al., 2023*; *DeLuca et al., 2020*). As egg follicles differentiate, nurse cells begin to accumulate H3K9me3 marks, and H3K27me3 histone marks begin to accumulate over more traditional polycomb domains. This in turn leads to a decrease in the number of open chromatin peaks as well as H3K27ac histone marks, which are generally associated with active transcription (*Pang et al., 2023*; *DeLuca et al., 2020*). Essentially, these data support the idea that egg chambers restrict gene expression competency as they differentiate.

In order to determine the relationship in our OVO ChIP data and other chromatin marks, we analyzed GSC H3K27ac, H3K27me3, H3K9me3, H3K4me3, and ATAC-seq data (*Pang et al., 2023*; *DeLuca et al., 2020*) with the same parameters used to establish significant OVO peaks in our OVO ChIP dataset. Our OVO ChIP data was from one day old ovaries, and we did not profile specific follicle stages. So we also analyzed 32 c (roughly stage 5 egg chambers) H3K27ac, H3K27me3, ATAC-seq, and 8 c H3K9me3 (32 c was not available) histone marks (*Pang et al., 2023*; *DeLuca et al., 2020*) to see if there were any stage specific differences in comparison to OVO DNA binding. We focused specifically on GSC and 32 c egg follicles for these chromatin marks since that is when the *ovo* hypomorphic egg chambers arrest (*Benner et al., 2023*). We first plotted the read density of each respective chromatin mark minus their input control centered on either the OVO ChIP peak local maximum (*Figure 2A*) or OVO DNA binding motifs contained within significant OVO ChIP peaks (*Figure 2B*).

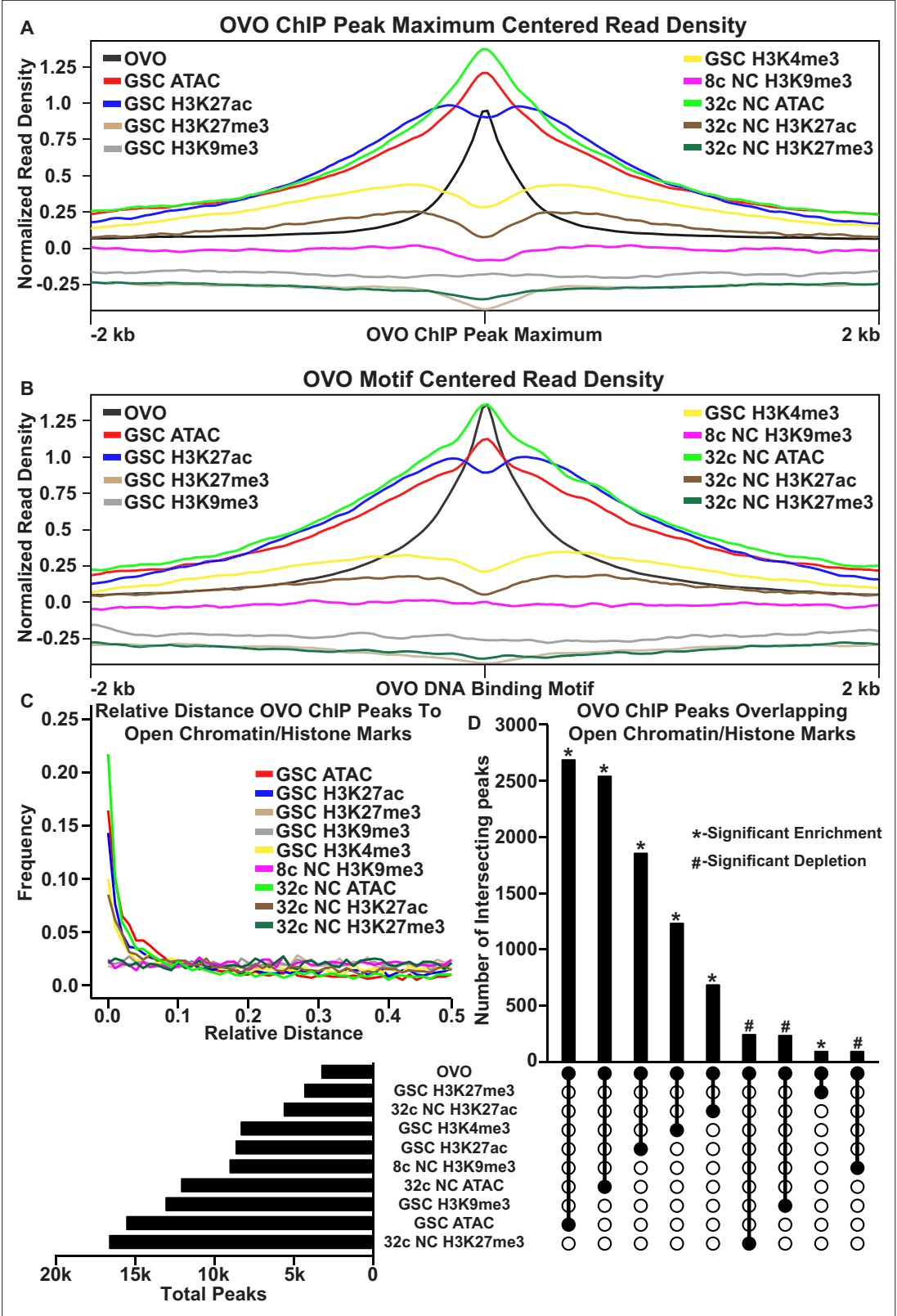

**Figure 2.** OVO DNA binding is associated with open chromatin and transcriptionally active histone marks. (**A, B**) OVO ChIP minus input control, GSC and 32 c ATAC-seq, GSC H3K27ac, H3K4me3, H3K27me3, H3K9me3, 8 c NC H3K9me3, 32 c NC H3K27ac, and H3K27me3 ChIP-seq read coverage density centered on each OVO peak maximum or OVO DNA binding motif located within a significant OVO ChIP peak. (**C**) Relative distance of OVO ChIP peaks to significantly called peaks for GSC and 32 c ATAC-seq, GSC H3K27ac, H3K4me3, H3K27me3, H3K9me3, 8 c NC H3K9me3, 32 c NC

*Figure 2 continued on next page*

*Figure 2 continued*

H3K27ac, and H3K27me3 ChIP-seq genome-wide. (**D**) Total number of significant peaks (left) and the total number of overlapping peaks (top) between OVO ChIP and GSC and 32 c ATAC-seq, GSC H3K27ac, H3K4me3, H3K27me3, H3K9me3, 8 c NC H3K9me3, 32 c NC H3K27ac, and H3K27me3 ChIP-seq. Lines connecting solid dots indicates the amount of overlapping peaks between those two corresponding datasets. Asterisk indicates significantly enriched overlap while hashtag indicates significantly depleted overlap between datasets.

The online version of this article includes the following figure supplement(s) for figure 2:

**Figure supplement 1.** OVO DNA binding is associated with open chromatin and transcriptionally active histone marks across variations of gene binding patterns.

GSC ATAC and H3K27ac read density showed a high degree of enrichment over OVO ChIP peak maximums (*Figure 2A*) and OVO DNA binding motifs (*Figure 2B*), consistent with positive transcriptional activity. GSC H3K4me3 read density was, to a lesser extent, also enriched with OVO ChIP peak maximums and OVO DNA binding motifs. However, there was no read density enrichment for repressive GSC H3K27me3 and H3K9me3 histone marks. The same findings generally held true when looking at the overlap of OVO ChIP peaks and chromatin marks in differentiating egg chambers. Notably, there was an even higher read density enrichment over OVO ChIP peak maximum and DNA binding motifs for 32 c ATAC-seq data, while read density enrichment decreased for 32 C H3K27ac histone mark (*Figure 2A and B*).

Since there was a high degree of read density enrichment between OVO ChIP and other chromatin marks/low nucleosome density, we wanted to determine the extent of the overlap between significant OVO ChIP peaks and significantly called peaks from the different types and stages of histone marks and ATAC-seq data. To do this, we measured the relative distance of OVO ChIP peaks to the same datasets described above. We found that OVO ChIP peaks had a lower relative distance, and thus were spatially overlapping/closer in the genome, to 32 c NC ATAC, GSC ATAC, GSC H3K27ac, GSC H3K4me3, and 32 c NC H3K27ac peaks, in that order (*Figure 2C*). While the relative distance between OVO ChIP peaks and H3K9me3 and H3K27me3, regardless of stage, showed no spatial association. There was also further support for this association with transcriptionally active histone marks/open chromatin when measuring the overlap between significant OVO ChIP peaks and the respective significant histone ChIP and ATAC-seq peaks (*Figure 2D*). A Fisher's exact test indicated a significant enrichment in overlapping peaks genome-wide between OVO and GSC ATAC (p<0.001, odds ratio = 75.9), 32 c NC ATAC (p<0.001, odds ratio = Infinite), GSC H3K27ac (p<0.001, odds ratio = 31.7), GSC H3K4me3 (p<0.001, odds ratio = 12.0), and 32 c NC H3K27ac (p<0.001, odds ratio = 7.9) peaks. While there was a significant depletion in overlapping peaks genome-wide between OVO and 32 c NC H3K27me3 (p<0.001, odds ratio = 0.6), GSC H3K9me3 (p<0.001, odds ratio = 0.7), and 8 c NC H3K9me3 (p<0.001, odds ratio = 0.5).

The association of OVO binding with active histone marks and open chromatin was striking, but open chromatin is likely a general phenomenon of promoters (*Haines and Eisen, 2018*). Indeed, when measuring the read density for GSC and 32 C ATAC-seq for OVO bound and OVO non-bound promoters, there is an enrichment for open chromatin at the TSS regardless of OVO binding. However, we did notice an increase in enrichment for OVO-bound promoters compared to OVO non-bound promoters (*Figure 1—figure supplement 1G*), possibly suggesting that OVO-bound promoters are more open or have an increase in accessibility when compared to non-OVO bound promoters. This same relationship held true for the transcriptionally active histone mark H3K27ac in GSCs (*Figure 1—figure supplement 1H*). Since only 45% of OVO ChIP peaks overlapped TSSs, we plotted the read density of the above chromatin marks over OVO ChIP peak maximums for OVO bound over the TSS, gene body, or intergenic regions (*Figure 2—figure supplement 1A–D*). We found that OVO-bound regions that were not overlapping the TSS still showed the same propensity for enrichment of open chromatin and active histone marks. Intergenic regions were especially enriched for open chromatin measured through ATAC-seq. Altogether suggesting that OVO binding genome-wide is tightly associated with open chromatin regardless of germ cell stage, and active transcription in GSCs. In other words, chromatin state data suggests OVO is acting positively on its target genes and raises the possibility that OVO-binding and open chromatin are related.

## OVO DNA binding motifs are evenly distributed around promoters and are enriched for INR, DPE, and MTE elements

Our data thus far clearly indicates that OVO binding occurs at or very near the core promoter, a region recognized by an enormous collection of factors that associate with RNA polymerase to initiate transcription (*Aoyagi and Wassarman, 2000*; *Ngoc et al., 2019*). The highly organized polymerase complex has sequence-specific DNA recognition sites with incredibly precise spacing between them, with an overall DNA footprint of a little less than 100 bp (*Rice et al., 1993*; *FitzGerald et al., 2006*; *Ohler et al., 2002*). There are upstream binding sites such as TATA, sites at transcription start, such as the initiator (INR), and downstream promoter elements (DPE; *Ngoc et al., 2019*). The combinations of these DNA motifs is not random in mammals and *Drosophila* (*FitzGerald et al., 2006*), and distinct combinations of different motifs at the TSS of genes expressed in *Drosophila* are conserved over tens of millions of years of evolution (*Chen et al., 2014*). The male germline expresses a number of TATA-associated factors that have been implicated in male-specific promoter usage for gene expression (*Hiller et al., 2004*; *Hiller et al., 2001*; *Lu et al., 2020*; *Li et al., 2009*). It is possible that OVO is a female germline specific TATA-associated factor, and if so, OVO-binding sites at core promoters should share precise spacing with other core promoter elements, suggesting it is likely part of the complex. If not, then OVO is more likely to facilitate binding of the basal transcriptional machinery. Because of the extended footprint of engaged RNA polymerase, OVO and the basal machinery would not be likely to occupy the same region at the same time.

Like OVO ChIP peaks, OVO DNA binding motifs were highly enriched at or near the TSS (*Figure 3A*). We carefully analyzed the spacing of these sites relative to core promoter elements to see if spacing was precise at the nucleotide level. We first searched for the presence of previously defined DNA motifs that are enriched at promoters (*FitzGerald et al., 2006*; *Ohler et al., 2002*; *Lim et al., 2004*) using FIMO (*Grant et al., 2011*). We defined promoters by using the DNA sequences 150 nucleotides upstream and downstream of the significant dominant TSSs in our previously analyzed ovary CAGE-seq datasets (*Chen et al., 2014*). After extracting these sequences and searching for significant scoring motifs, we plotted the density of each motif in relation to the empirically mapped TSSs (*Figure 3B*). We also searched for all OVO motifs found in our significant ChIP peaks within these promoter sequences. When plotting the density of DNA motifs found in ovary CAGE-seq promoters, we found that there were prominent peaks for INR and M1BP (M1BP *Li and Gilmour, 2013*)=Ohler 1 (*Ohler et al., 2002*)=DMv4 (*FitzGerald et al., 2006*) near the TSS, and MTE (*Lim et al., 2004*) and DPE elements downstream of the TSS. This distribution and frequency are consistent with the constrained location of these DNA motifs (*FitzGerald et al., 2006*; *Ohler et al., 2002*; *Chen et al., 2014*). Significantly, the OVO DNA binding motifs showed a broad distribution upstream and downstream of the TSS.

The precise core promoter architecture of OVO-bound TSSs is revealed in the CAGE-seq dataset. Plotting the distribution of classical core promoter sequence elements in OVO-bound promoters showed a similar, but exaggerated, profile compared to all core promoters of the ovary CAGE-seq dataset. We found a significant enrichment for INR (p<0.01, odds ratio = 1.70), DPE (p<0.01, odds ratio = 1.81), MTE (p<0.01, odds ratio = 1.65), and most importantly, OVO DNA binding motifs (p<0.01, odds ratio = 4.83), in ovary promoters that overlapped an OVO ChIP peak in comparison to the subset of ovary promoters that did not overlap an OVO ChIP peak (*Figure 3C and D*). This indicates that OVO-bound promoters are more likely to contain these specific promoter elements than non-OVO-bound promoters. As has been described before, promoters containing INR and DPE, but lacking TATA-box elements, are common among *Drosophila* gonad promoters compared to promoters of other tissue types (*Chen et al., 2014*). The presence of TATA-box elements is negatively associated with germline-specific gene expression (*FitzGerald et al., 2006*). We found that TATA-box elements were significantly depleted in ovary CAGE-seq promoters when compared to testes (p<0.01, odds ratio = 0.78; *Figure 3E*) or digestive system (p<0.01, odds ratio = 0.50; *Figure 3F*) CAGE-seq promoters. Indeed, both *ovo* and *otu* have TATA-less promoters. Briefly, OVO bound promoters are characterized by the presence of INR, DPE, MTE, and, of course, OVO DNA binding motifs. This could represent a functional class of promoters utilized for gene expression in the *Drosophila* ovary. Importantly, the distribution of OVO DNA binding motifs in ovary promoters is not fixed relative to TSSs or other core promoter elements. Thus, it is highly unlikely that OVO acts as a female germline RNA polymerase complex member that anchors the complex to the core promoter and helps determine

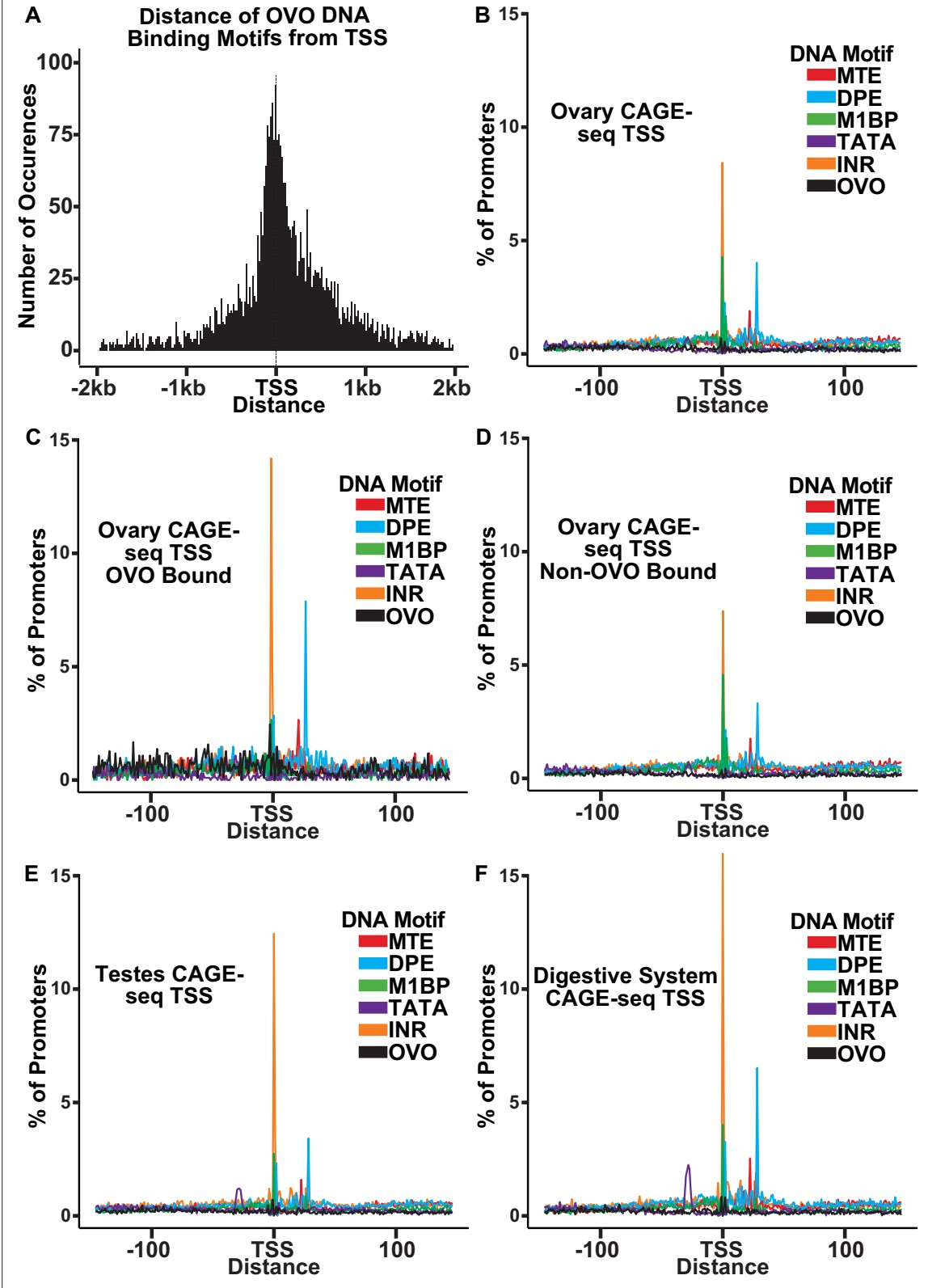

**Figure 3.** OVO bound promoters are enriched for INR, DPE, and MTE elements. (**A**) Histogram of the distance of in vivo and in vitro OVO DNA binding motifs within significant overlapping OVO ChIP peaks from the closest genes TSS. (**B–F**) Histogram of the percent of promoters from tissue-specific CAGE-seq analysis of common promoter motif elements centered on the dominant significant TSS.

the + 1 mRNA nucleotide. Rather, the imprecise location of OVO-binding sites might suggest that OVO is more likely to facilitate the binding of other basal transcriptional factors.

## OVO activates gene expression in the female germline

Occupancy is a requirement for activity, but occupancy does not equal activity. Understanding the transcriptional consequences of OVO occupancy genome-wide would allow us to investigate mechanisms. However, as we mentioned earlier, the fact that *ovo* is absolutely required for female germline viability greatly complicates this analysis. Measuring gene expression in dead or dying germ cells was unlikely to be informative. Previous work from our lab has identified a transheterozygous *ovo* allelic combination (*ovo*$^{ovo-GAL4}$/*ovo*$^{ΔBP}$) that greatly reduces OVO activity resulting in sterility, however, female germ cells are able to survive up until at least stage 5 of oogenesis (*Benner et al., 2023*). *ovo*$^{ovo-GAL4}$ is a CRISPR/Cas9 derived T2A-GAL4-3xSTOP insertion upstream of the splice junction of exon 3 in the *ovo-RA* transcript (*Figure 1—figure supplement 1A*). Importantly, this insertion in the extended exon 3 would disrupt roughly 90% of the *ovo-B* transcripts. However, since about 10% of *ovo-B* transcripts utilize an upstream splice junction in exon 3, these transcripts would not be disrupted with the T2A-GAL4-3xSTOP insertion and thus allow for enough OVO activity for germ cell survival (*Benner et al., 2023*). Since *ovo*$^{ovo-GAL4}$ expresses GAL4 in place of full-length OVO due to the T2A sequences, we can drive expression of a rescuing OVO-B construct downstream of *UASp* to generate OVO$^+$ female germ cells, which in fact does rescue the arrested germ cell phenotype of *ovo*$^{ovo-GAL4}$/*ovo*$^{ΔBP}$ ovaries. Therefore, in order to determine genes that are transcriptionally responsive to OVO, we compared the gene expression profiles in sets of ovaries that had the *ovo* hypomorphic phenotype with a negative control rescue construct (*ovo*$^{ovo-GAL4}$/*ovo*$^{ΔBP}$; *UASp-GFP*)(*Figure 4A*) versus those that drive expression of the rescue construct expressing OVO-B (*ovo*$^{ovo-GAL4}$/*ovo*$^{ΔBP}$; *UASp-3xFHA-OVO-B*)(*Figure 4B*).

Since *ovo*$^{ovo-GAL4}$/*ovo*$^{ΔBP}$; *UASp-3xFHA-OVO-B* females have full rescue of the arrested germ cell phenotype seen in *ovo*$^{ovo-GAL4}$/*ovo*$^{ΔBP}$; *UASp-GFP* females, we needed to take further measures to ensure our analysis of gene expression was stage comparable between the two sets of ovaries. The adult female ovary contains somatic cells, germline stem cells, and germline derived nurse cells that would be profiled in a bulk ovary tissue RNA-seq experiment. Although OVO is only required and expressed in germline derived cell types, we chose to dissect 1-day-old post-eclosion *ovo*$^{ovo-GAL4}$/*ovo*$^{ΔBP}$; *UASp-3xFHA-OVO-B* female ovaries to enrich for early stages of oogenesis and collected only ovarioles containing the germarium through previtellogenic egg chambers. *ovo*$^{ovo-GAL4}$/*ovo*$^{ΔBP}$; *UASp-GFP* ovaries were collected at the same age post-eclosion and we specifically collected ovaries that contained a visible ovariole structure (and therefore contained germ cells) to minimize comparing germ cells to somatic ovary structures, but rather germ cells to germ cells. We then performed RNA-seq in quadruplicate and measured the changes in gene expression between ectopic rescue OVO and hypomorphic OVO ovaries. We used a significance level of p-adj <0.05 and a log2 fold change cutoff of >|0.5| to call differential expression between these two sets of ovaries. We utilized these log2 fold change cutoffs for two reasons. Our control ovary genotype (*ovo*$^{ovo-GAL4}$/*ovo*$^{ΔBP}$; *UASp-GFP*) has hypomorphic OVO activity, hence germ cells can survive but are arrested. With the addition of ectopic rescue OVO in *ovo*$^{ovo-GAL4}$/*ovo*$^{ΔBP}$; *UASp-3xFHA-OVO-B* ovaries, we predicted that genes that were directly regulated by OVO would transcriptionally respond, however, we were unsure as to what degree the response would be in comparison to hypomorphic OVO. We reasoned that if the changes were not significant between genotypes, then minor changes in gene expression would not matter. Our second reason for using these cutoffs is we had an internal control between the two genotypes. We knew through immunostaining that Vas protein was present in the germline of both genotypes (*Figure 4A and B*) and therefore was likely expressed at similar levels in the germline of both genotypes. Both genotypes also expressed *GAL4* under the control of *ovo* in the germline. We examined the expression levels of *vas* and *GAL4* and found that *vas* had a log2 fold change of 0.15 (p-adj=0.03) and *GAL4* had a log2 fold change of 0.33 (p-adj=0.18) (*Figure 4C*). These data suggest a slight underrepresentation of germline expression in *ovo* hypomorphic ovaries. Therefore, by using the greater than 0.5 and less than –0.5 log2 fold change cutoffs, and a less than 0.05 p-adj value cutoff, we would be conservative to not call genes differentially expressed due to differences in the relative abundance of germ cells and somatic cells.

We were able to reliably detect the expression of 10,804 genes in these early ovarioles (*Supplementary file 4*). The differential expression analysis indicated that 1994 genes primarily expressed

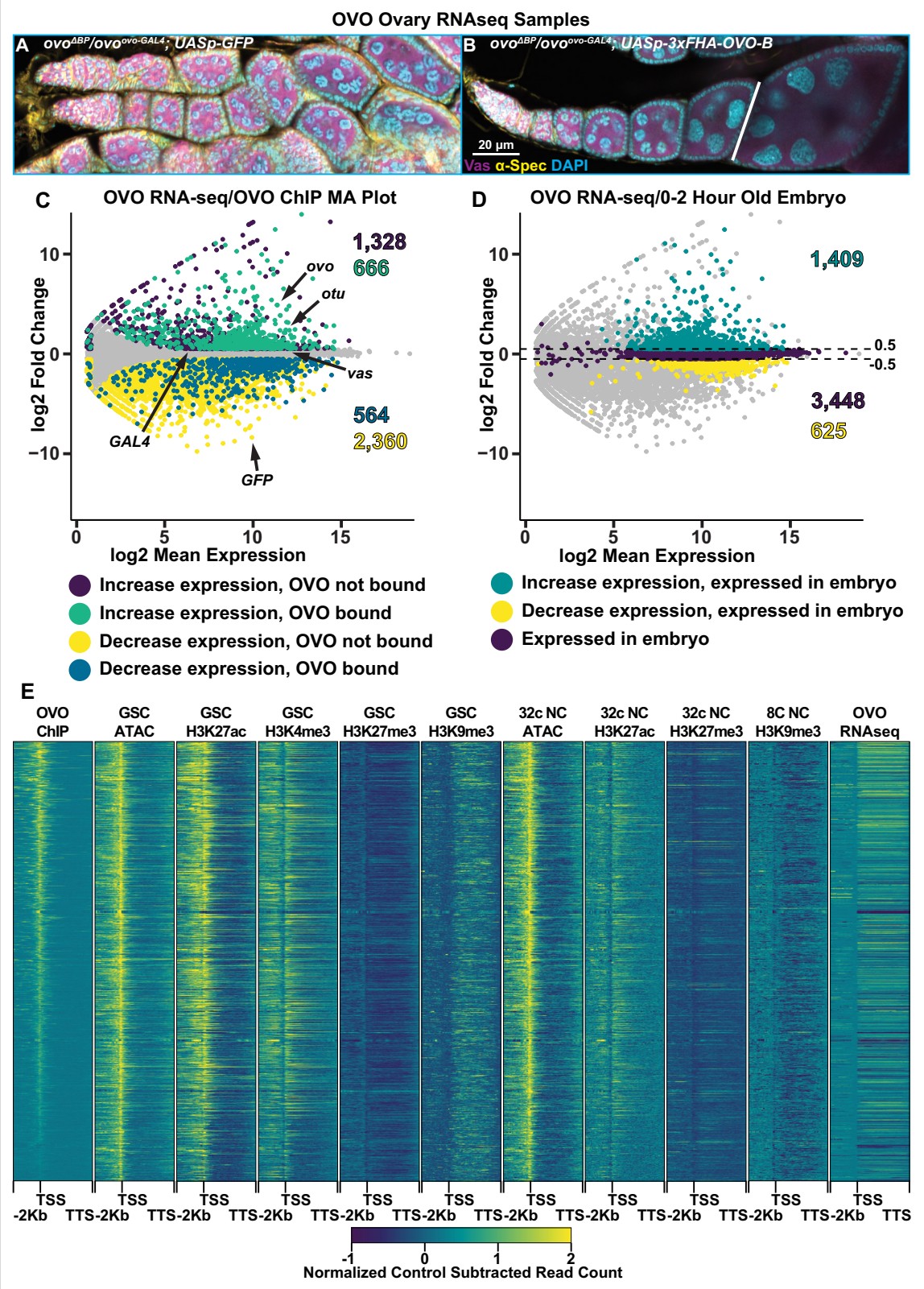

**Figure 4.** Genes bound by OVO increase in expression in the presence of OVO genome-wide. (**A, B**) Immunofluorescent staining of adult ovarioles of the indicated genotypes (20 x, scale bar = 20 μm). Ovarioles were stained for Vas (purple) to label the germline, α-Spectrin (yellow) to label dot spectrosome and fusomes, and DAPI (cyan) to label nuclei. Line indicates the dissection point for germarium through previtellogenic RNA-seq samples. (**C**) MA plot of *ovo^ΔBP^/ovo^ovo-GAL4^; UASp-3xFHA-OVO-B* versus *ovo^ΔBP^/ovo^ovo-GAL4^; UASp-GFP* RNA-seq differential expression results. Purple dots indicate

*Figure 4 continued on next page*

*Figure 4 continued*

genes that significantly increased in expression and were not bound by OVO, cyan dots indicate genes that significantly increased in expression and were bound by OVO, yellow dots indicate genes that significantly decreased in gene expression and were not bound by OVO, blue dots indicate genes that significantly decreased in gene expression and were bound by OVO, and gray dots indicate genes that were not differentially expressed from our analysis. (D) MA plot of *ovo^ΔBP^/ovo^ovo-GAL4^; UASp-3xFHA-OVO-B* versus *ovo^ΔBP^/ovo^ovo-GAL4^; UASp-GFP* RNA-seq differential expression results. Cyan dots indicate genes that significantly increased in expression and were found to be moderately expressed in 0–2 hr old embryos, yellow dots indicate genes that significantly decreased in gene expression and were found to be moderately expressed in 0–2 hr old embryos, purple dots indicate genes that were not differentially expressed and were found to be moderately expressed in 0–2 hr old embryos, and gray dots indicate genes that were not differentially expressed and were not found to be moderately expressed in 0–2 hr old embryos. (E) Gene level read coverage heatmaps of OVO ChIP minus input, GSC and 32 c ATAC-seq, GSC H3K27ac, H3K4me3, H3K27me3, H3K9me3, 8 c NC H3K9me3, 32 c NC H3K27ac, and H3K27me3 ChIP-seq, and *ovo^ΔBP^/ovo^ovo-GAL4^; UASp-3xFHA-OVO-B* minus *ovo^ΔBP^/ovo^ovo-GAL4^; UASp-GFP* RNA-seq for genes bound by OVO. The order of the heatmap is genes with the highest to lowest amount of OVO ChIP read density.

in the germline (see next paragraph) significantly increased in expression with ectopic rescue OVO (*Figure 4C*, cyan/purple dots) and 2924 genes primarily expressed in the soma (see next paragraph) significantly decreased in expression with ectopic rescue OVO expression (*Figure 4C*, yellow/blue dots). A total of 5886 genes were not considered to be differentially expressed in our analysis (*Figure 4C*, gray dots). OVO is expressed in the germline, not the soma, and previous work has shown that OVO-B is a transcriptional activator (*Andrews et al., 2000*), so we hypothesized that many of the genes increasing in expression in the presence of rescuing OVO were direct downstream targets. We found that 2,298 genes that were expressed in our RNA-seq data overlapped an OVO ChIP peak. 666 genes significantly increased in expression and were bound by OVO, which is a significant enrichment according to a Fisher's exact test (*Figure 4C*, cyan dots, p<0.01, odds ratio = 2.21). While conversely, 564 genes decreased in expression and were bound by OVO, indicating a significant depletion according to a Fisher's exact test (*Figure 4C*, blue dots, p<0.01, odds ratio = 0.85). This strongly suggests that genes that are bound by OVO, transcriptionally respond in a positive manner. This finding is fully consistent with our meta-analysis comparing OVO ChIP-seq and histone ChIP/ATAC-seq data (*Figure 4E*). OVO binding was highly associated with transcriptionally active histone marks such as H3K27ac and H3K4me3, open chromatin, and increased expression.

There are genes that showed decreased expression in the OVO rescued ovaries, but we believe this is technical rather than biological. OVO is expressed only in the germline, but ovarioles contain germ cells and somatic cells. The presence of empty ovarioles, containing leftover strings of somatic cells, are evident even in *ovo^ovo-GAL4^/ovo^ΔBP^; UASp-GFP* ovaries that contain germ cells. Conversely, *ovo^ovo-GAL4^/ovo^ΔBP^; UASp-3xFHA-OVO-B* ovaries are fully rescued, and therefore possess more germ cell containing ovarioles than *ovo^ovo-GAL4^/ovo^ΔBP^; UASp-GFP* ovaries (*Benner et al., 2023*). Despite our best efforts to dissect individual ovarioles with a full complement of germ cells and egg chambers, we wondered if there might be fewer germ cells and egg chambers in the *ovo^ovo-GAL4^/ovo^ΔBP^; UASp-GFP* ovaries. To confirm that genes increasing in expression in ectopic rescue OVO were germline derived, we cross-referenced the significantly expressed genes in our RNA-seq datasets with the modENCODE developmental RNA-seq datasets (*Graveley, 2010*). We extracted the gene names of all genes that were considered to be 'moderately expressed' in 0–2 hr old embryos, which are produced during oogenesis and are deposited into the early embryo. We found that 71% of genes (1409/1994) that had a significant increase in expression in the presence of ectopic rescue OVO were found to be expressed in 0–2 hr old embryos (*Figure 4D*, green dots), while only 21% of genes (625/2,924) that had a significant decrease in expression were found in the same embryo dataset (*Figure 4D*, yellow dots). A total of 3448 genes from the 0–2 hr-old embryo dataset were not differentially expressed in our RNA-seq dataset (*Figure 4D*, purple dots). A Fisher's exact test confirmed that there was a significant enrichment for genes that significantly increased in expression and were present in 0–2 hr old embryos (p<0.01, odds ratio = 2.8). In comparison, there was a significant depletion for genes that significantly decreased in expression and were present in 0–2 hr old embryos (p<0.01, odds ratio = 0.17). This result indicated that genes that significantly increased in expression were more likely to be expressed in the germline and that the presence of ectopic rescue OVO significantly increased the expression of genes that were maternally deposited in the early embryo. While the set of genes that significantly decrease in expression are not enriched in the embryo and are more likely specific to somatic cell gene expression. These genes are unlikely to be direct OVO targets due to the absence of OVO in those cells, although we certainly cannot rule out the possibility of a non-autonomous effect

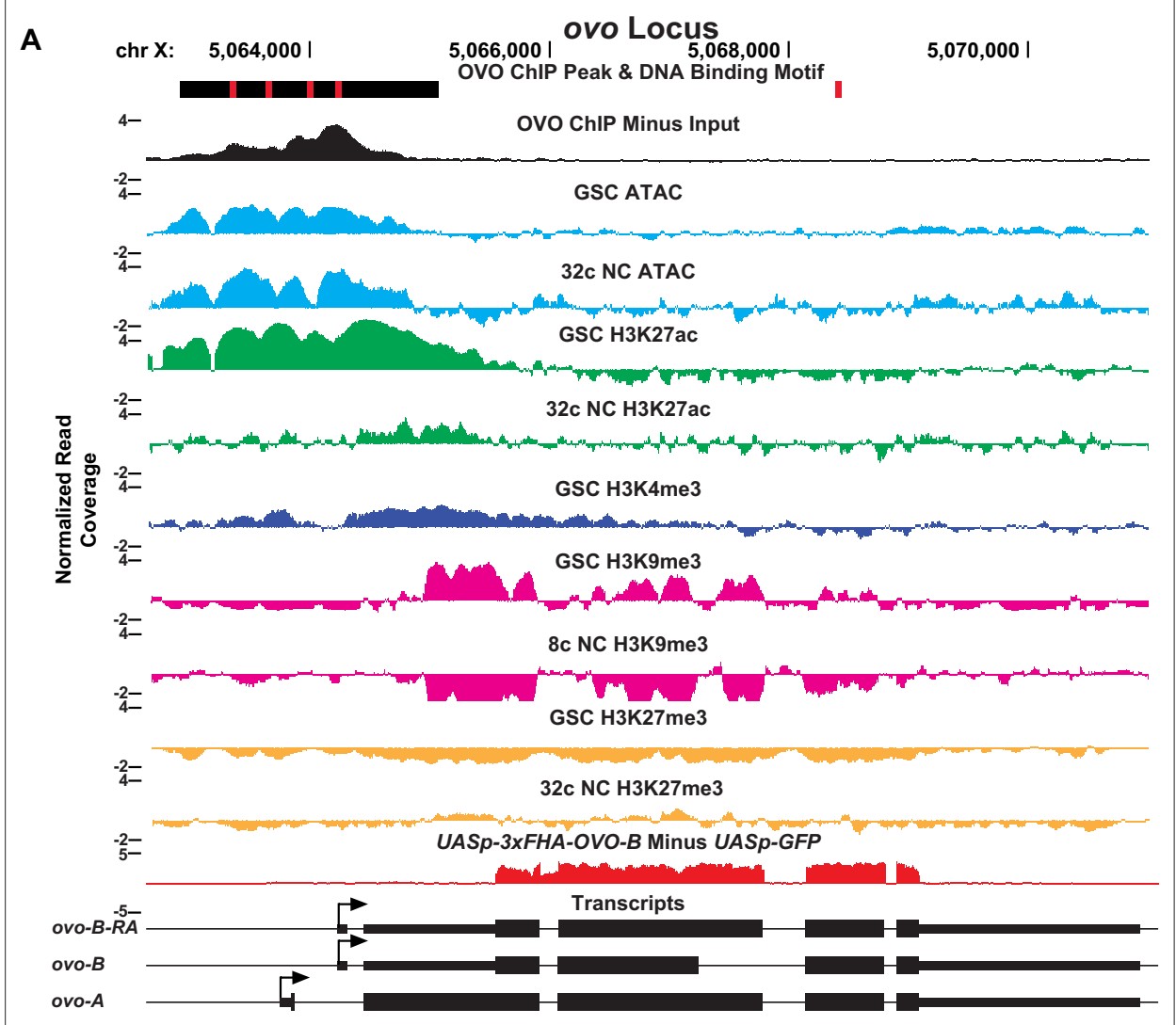

**Figure 5.** OVO ChIP-seq, ATAC/histone ChIP-seq, RNA-seq, and DNA binding motifs at the *ovo* locus. (**A**) *ovo* gene level read coverage tracks for OVO ChIP minus input (black), GSC and 32 c ATAC-seq (light blue), GSC and 32 C H3K27ac (green), H3K4me3 (dark blue), GSC and 32 c H3K27me3 (orange), and GSC and 8 c H3K9me3 (pink) ChIP-seq, and *ovo^{ΔBP}*/*ovo^{ovo-GAL4}*; *UASp-3xFHA-OVO-B* minus *ovo^{ΔBP}*/*ovo^{ovo-GAL4}*; *UASp-GFP* RNA-seq (red). Red rectangles and black rectangles represent significant OVO DNA binding motifs and OVO ChIP peaks, respectively. Gene models are represented at bottom. Small rectangles represent untranslated regions, large rectangles represent translated regions. Arrows indicate transcriptional start sites.

The online version of this article includes the following figure supplement(s) for figure 5:

**Figure supplement 1.** OVO ChIP-seq, ATAC/histone ChIP-seq, RNA-seq, and DNA binding motifs at the *otu* locus.

of OVO on somatic gene expression. In terms of the germline proper, OVO appears to be a positively acting transcription factor.

## OVO positively regulates essential oogenesis genes

We wanted to examine a subset of the OVO target genes in detail, and began with the known OVO targets, *ovo* itself and *otu* (***Bielinska et al., 2005***; ***Lü et al., 1998***; ***Lü and Oliver, 2001***). Since the relationship between OVO binding to these two genes has been well-characterized, we validated the OVO ChIP, histone ChIP/ATAC-seq, and RNA-seq datasets by examining these two genes first. Since OVO positively regulates the expression of both these genes, then we would expect OVO to be physically bound at OVO motifs required for high transcription, the presence of transcriptionally active histone marks and open chromatin at these loci, as well as a positive transcriptional response in the presence of rescuing OVO-B. This is exactly what we observed. A significant OVO ChIP peak

was found overlapping the TSS of *ovo-B*, with four significant OVO DNA binding motifs present (*Figure 5A*). ATAC-seq, H3K27ac, and H3K4me3 peaks overlapped the *ovo* promoter. Transcriptionally, *ovo* RNA-seq reads are likely derived from the *UASp-3xFHA-OVO-B* cDNA rescue or are indistinguishable between the genomic locus and rescuing cDNA transgene. We found a non-significant increase in exon 3 to exon 4 intronic *ovo* reads with the expression of ectopic rescue OVO (log2 fold change = 0.76, p-adj=0.26). These intronic reads would be derived from the endogenous *ovo* locus, but it is difficult to conclusively determine if the endogenous *ovo* locus would respond transcriptionally to ectopic OVO downstream of *UASp* (for example, the pathway for *ovo* is no longer autoregulatory in *ovo*^*ovo-GAL4*^/*ovo*^*ΔBP*^; *UASp-3xFHA-OVO-B* germ cells, there is an additional GAL4 >*UASp* activation step). So, we could not confidently assess whether *ovo* responded transcriptionally to ectopic rescue OVO. However, when looking at the *otu* locus (*Figure 5—figure supplement 1A*), we found OVO occupancy over the TSS of both annotated *otu* promoters, with significant OVO DNA binding motifs overlapping and in close proximity to the TSSs. The *otu* locus contained similar ATAC-seq and activating histone mark peaks overlapping the TSS found at the *ovo* locus. It was also evident that *otu* had a positive transcriptional response to the presence of OVO rescue (log2 fold change = 2.41; p-adj <0.001). These results confirm that OVO binds and positively regulates both itself and *otu* in vivo, as previous work has indicated.

Since our overlapping OVO ChIP-seq and RNA-seq data suggests that hundreds of genes that are bound by OVO increase in expression in the presence of ectopic rescue OVO, we wanted to know more about the functions of those genes. To do this, we performed Gene Ontology enrichment analysis with gProfiler software (*Raudvere et al., 2019*). To be especially stringent, we focused on the genes that contained an OVO ChIP peak overlapping the transcriptional start site and significantly increased in expression in the presence of rescue OVO. A total of 525 genes met these criteria. Biological process GO term enrichment analysis on these 525 genes showed a significant enrichment for 45 GO terms (*Supplementary file 5*). The significant GO terms were almost exclusively related to female reproduction and maternal control of early embryonic development (*Figure 6A*). Genes that are required for processes such embryonic axis specification, mRNA localization, egg activation, and translational regulation were found within these significantly enriched GO terms. These GO terms are well understood in the context of oogenesis and broadly suggest that OVO expression in adult gonads is essential for constructing an egg and depositing maternal RNAs to support zygotic embryonic development.

GO term enrichment analysis of genes that are bound by OVO and increase in expression in the presence of ectopic rescue OVO suggested that OVO is likely a main transcriptional regulator of oogenesis. These genes are the subject of decades of work on *Drosophila* oogenesis, but essentially all the work on them has focused on what they do, not on how they are transcriptionally regulated. For example, *bicoid* (*Figure 6B*), and *bicoid* mRNA binding proteins *exuperantia* (*exu*) and *swallow* (*swa*), are essential for anterior specification of the embryo (*Lasko, 2012*). All these genes were occupied by OVO in vivo, significantly upregulated by OVO-B, and had OVO motifs in close proximity to the TSS. Genes involved in posterior patterning (*oskar* and *nanos*; *Figure 6C*), as well as pole cell specification genes (*polar granule component*, *germ cell-less*, and *aubergine*; *Lasko, 2012*; *Benner et al., 2018*), also showed similar RNA-seq, ChIP-seq, and OVO DNA binding motif profiles as *ovo* and *otu*. Genes that are involved in translational silencing and regulation of maternally provided mRNAs, such as *cup* (*Figure 6D*), *maternal expression at 31B* (*me31B*), *oo18 RNA-binding protein* (*orb*), and *bruno 1* (*bru1*), as well as essential genes involved in meiosis completion and egg activation after fertilization (*giant nuclei* (*gnu*), *pan gu* (*png*), *plutonium* (*plu*), *wispy* (*wisp*), *C(3)G*, and *matrimony* (*mtrm*)) (*Figure 6E*; *Lasko, 2012*; *Avilés-Pagán and Orr-Weaver, 2018*) all show this stereotypic pattern of promoter proximal OVO occupancy and DNA binding motifs, and OVO-dependent transcription. These data indicate that the OVO is a central transcription factor activating the expression of essential maternal and early embryonic development pathways in the female germline.

We also found that the genes *fs(1)N*, *fs(1)M3*, and *closca*, were all bound by OVO and responded transcriptionally to the presence of ectopic rescue OVO. These genes are significant because they constitute a set of genes that are expressed in the germline and the encoded proteins are eventually incorporated into the vitelline membrane providing the structural integrity and impermeability of the egg (*Mineo et al., 2017*; *Ventura et al., 2010*). Loss-of-function of these three genes results in flaccid eggs that are permeable to dye and fail to develop. The loss-of-function phenotype of *fs(1)N*, *fs(1)*

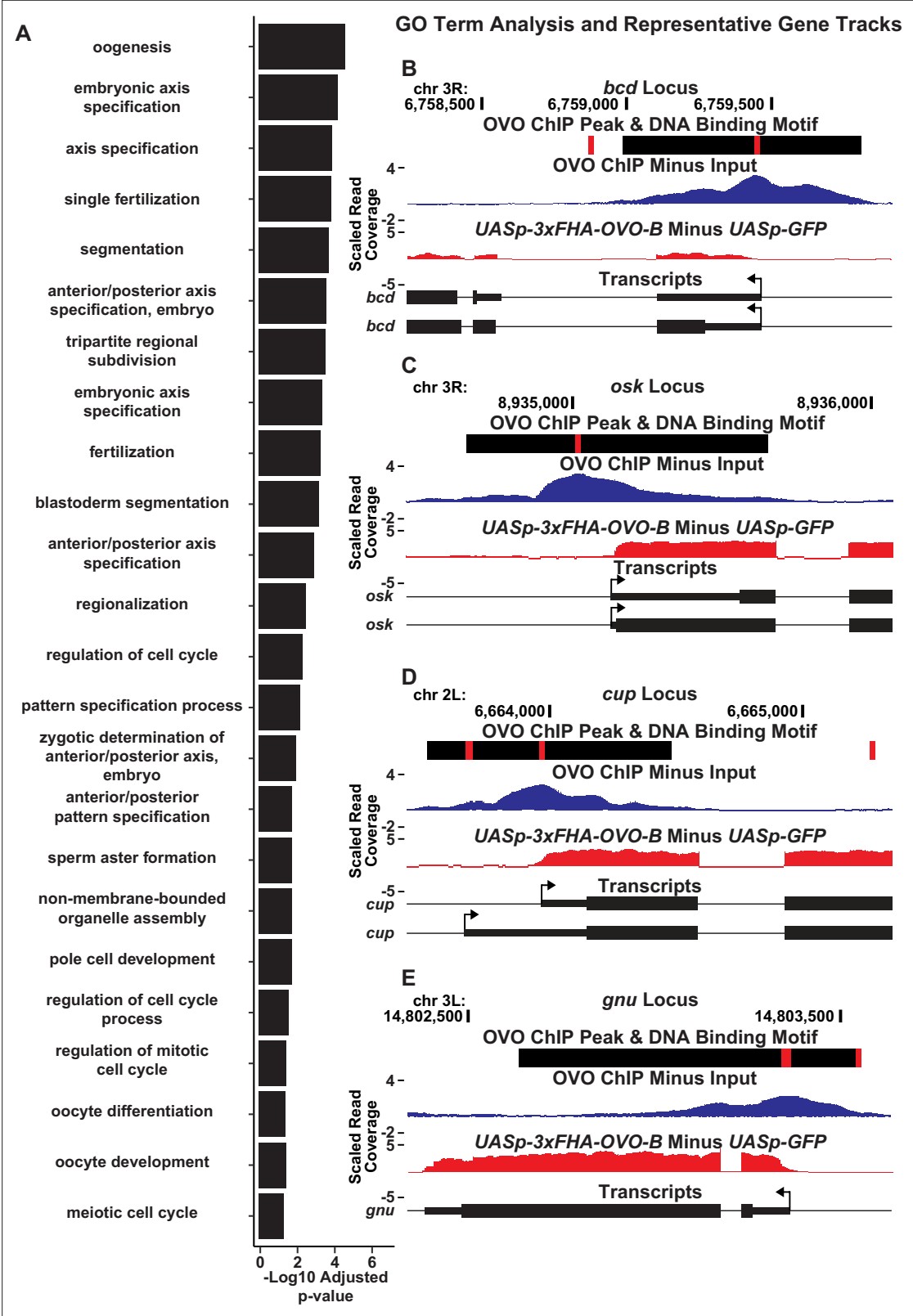

**Figure 6.** OVO binds and significantly increases the expression of a number of genes involved in essential maternal processes. (**A**) Significantly enriched GO biological process terms for genes bound by OVO and significantly increase in expression in the presence of ectopic rescue OVO. GO terms are restricted to the 24 smallest term sizes. (**B–E**) Example GO term gene level read coverage tracks for OVO ChIP minus input and *ovo*^ΔBP^/*ovo*^*ovo-GAL4*^; *UASp-3xFHA-OVO-B* minus *ovo*^ΔBP^/*ovo*^*ovo-GAL4*^; *UASp-GFP*. Red rectangles and black rectangles represent significant OVO DNA binding motifs and OVO ChIP

*Figure 6 continued*

peaks, respectively. Gene models are represented at bottom. Small rectangles represent untranslated regions, large rectangles represent translated regions. Arrows indicate transcriptional start sites.

*M3*, and *closca* closely resembles the dominant antimorph *ovo*^D3 phenotype. The *ovo*^D3 allele is the weakest of the original dominant-negative *ovo* alleles and produces defective eggs allowing us to explore the role of OVO in late stages (*Busson et al., 1983*; *Komitopoulou et al., 1983*). *ovo*^D3/ovo+ transheterozygous females express a repressive form of OVO that results in dominant sterility, and importantly, these females lay flaccid eggs with compromised vitelline membranes that are permeable to the dye neutral red (*Oliver et al., 1990*). Since OVO+ is bound at the TSS of *fs(1)N*, *fs(1)M3*, and *closca*, and these three genes respond transcriptionally to OVO+, then it is plausible that the repressive OVO^D3 is negatively regulating these three genes that are required for vitelline membrane formation. This is evidence that OVO is not only involved in regulating the expression of numerous essential maternal pathways for embryonic development, but it is also essential for regulating genes that are required for egg integrity and maturation.

## Discussion

Since its original isolation as a dominant female sterile locus in *Drosophila* (*Busson et al., 1983*; *Komitopoulou et al., 1983*), *ovo* has long been known as an essential gene in oogenesis. Female germ cells require *ovo* for survival and differentiation, while it has no described roles or functions in the male germline (*Oliver et al., 1987*; *Oliver et al., 1990*). OVO has also been found to be eternally present in the female germline, attesting to its likely continual requirement for female germ cell viability and identity (*Benner et al., 2023*). Our work here significantly expands our knowledge on OVO function in the female germline, showing that OVO binds and positively regulates a large array of genes required to build an egg and pattern the resulting embryo after fertilization. OVO accomplishes this by directly binding to the promoters of its targets, as well as more distant sites that could represent enhancers. Altogether, we suggest that OVO is a master transcriptional regulator coordinating a number of essential maternal pathways involved in oocyte and early embryonic development. Hints of these functions can be found in the hypomorphic and antimorphic *ovo* alleles which show egg chamber arrest, ventralized eggs, and permeable vitelline membranes. It is clear that OVO is required to activate multiple pathways involved in oocyte and early embryonic development.

The GO term enrichment analysis on genes that were bound by OVO and transcriptionally responded to OVO surprisingly indicated a large degree of overlap in oocyte and early embryonic developmental pathways. Also, OVO seemed to reinforce these pathways at multiple key genes within each pathway. For example, OVO bound to the promoters and increased the expression of *bcd*, and the *bcd* mRNA binding proteins *exu* and *swa*, all involved in ensuring correct anterior specification of the embryo (*Lasko, 2012*). Genes that are essential for egg activation were coordinately regulated by OVO as well. OVO downstream target genes *gnu*, *png*, *plu*, and *wisp* all belong in the same interconnected pathway ensuring egg activation (*Avilés-Pagán and Orr-Weaver, 2018*). A similar story was found for genes such as *cup*, *Me31b*, *bru1*, and *orb*, indicating that OVO controls a battery of genes involved in the positive regulation of RNA binding proteins that negatively regulate translation (*Lasko, 2012*). OVO also bound and positively regulated a number of posterior and germ plasm specification genes such as *osk*, *nos*, *aub*, *gcl*, and *pgc* (*Lasko, 2012*; *Mahowald, 2001*). Given this plethora of famous maternal effect loci, it might be tempting to suggest that OVO is sufficient for egg production, but there are important exceptions. For example, other important germ plasm factors such as *staufen* and *tudor* were not bound by or transcriptionally responsive to OVO. This observation suggest that other transcription factors are responsible for regulating these genes.

OVO binds in close proximity to the TSS of genes it positively regulates; however, it is still unclear precisely how it regulates gene expression. The possibilities include integration into the RNA polymerase complex itself, a short distance sigma factor like function, a core promoter conditioning function (pioneering), and/or garden variety transcription factor. Although active core promoters specifically in the ovary are enriched for OVO DNA binding motifs, we did not find a strict spatial orientation for these motifs in relation to the TSS, such as is found with other DNA elements such as INR, DPE, and MTE (*Ohler et al., 2002*; *FitzGerald et al., 2006*; *Lim et al., 2004*). It is therefore unlikely that OVO

is a core component of the RNA polymerase complex in the female germline. This suggests that it is unlikely to be analogous to male specific TATA-associated factors that have been shown to activate gene expression in the male germline (*Hiller et al., 2004*; *Hiller et al., 2001*; *Lu et al., 2020*; *Li et al., 2009*). It is therefore possible, and previously well supported, that OVO is an activator of transcription (*Lü et al., 1998*; *Lü and Oliver, 2001*; *Bielinska et al., 2005*). One aspect of OVO DNA binding that showed differences with stage-specific histone ChIP and ATAC-seq, was OVO's strong association with open chromatin. The role of repressors of transcription such as the polycomb complex, *egg*, *wde*, and *Su(var)205* in restricting gene expression through promoting heterochromatin formation in differentiating egg chambers is well established (*Smolko et al., 2018*; *DeLuca et al., 2020*). OVO might ensure that the chromatin status of maternally expressed genes remains open. Evidence from our work points to OVO fulfilling that role.

In GSCs, OVO ChIP peaks largely overlap open chromatin and transcriptionally active histone marks. However, in stage 5 egg chambers, there was an even higher degree of association with open chromatin (almost all OVO ChIP peaks overlapped ATAC-seq peaks), while the significant association with H3K27ac marks was greatly reduced. This difference is likely significant. As GSCs differentiate they accumulate repressive chromatin marks while the number of ATAC-seq and H3K27ac peaks are reduced. This increase in association with open chromatin and OVO binding, even as the amount of open chromatin is reduced throughout egg chamber differentiation, might indicate that OVO binding helps to maintain chromatin accessibility, even when the locus in question is no longer actively transcribed. The loss of histone marks of active transcription at OVO-bound open chromatin in later differentiating egg chambers might mean that OVO does not influence the transcriptional potential of target genes as strongly as it influences the chromatin status in this second phase. Therefore, OVO might be more similar in function to pioneer factors/chromatin remodelers than it is to a transcription factor that is only involved in activating transcription.

The requirement for OVO at the TSS of target genes has been well characterized at its own locus as well as its downstream target *otu*. Our OVO ChIP and expression data confirm findings from previous work that OVO is binding to these target promoters, and in the case of *otu*, strongly responds transcriptionally to the presence of OVO. Although we did not test the requirement for OVO DNA binding motifs at other OVO-bound genes in this work, this has been extensively explored before, showing that removal of OVO DNA binding sites overlapping the TSS results in a strong decrease in reporter expression (*Lü et al., 1998*; *Bielinska et al., 2005*; *Lü and Oliver, 2001*). Removal of more distal upstream OVO DNA binding sites also reduces reporter expression to a lesser degree. However, for most cases tested, removal of OVO DNA binding sites while leaving the rest of the enhancer regions intact, never totally abolished reporter expression. These dynamics are highly similar to work that has been completed on the pioneer factor *zelda* (*zld*). Adding *zld* DNA binding motifs to a stochastically expressed transcriptional reporter increases the activity and response of the reporter (*Dufourt et al., 2018*). Distally located *zld* DNA binding motifs influenced reporter expression to a lesser degree than proximal sites. A single *zld* DNA binding site adjacent to the TSS produced the strongest reporter activity. Importantly, just like the activity of OVO transgenic reporters, there is not an absolute requirement for *zld* DNA binding to activate reporter expression; however, the addition of TSS adjacent *zld* DNA binding motifs does strongly influence reporter response. We know that *zld* achieves this reporter response through its pioneering activity (*Xu et al., 2014*; *Harrison et al., 2011*), whether OVO achieves this similar effect on gene expression through a shared mechanism, or in cooperation with other transcription factors needs to be further explored.

## Methods

All reagents used in this study can be found in the FlyBase recommended supplementary ART table (*Supplementary file 6*).

### Fly husbandry

All fly crosses were conducted at 25 °C with 65% relative humidity and constant light unless otherwise noted. Flyfood consisted of premade flyfood (D20302Y) from Archon Scientific (Durham, NC).

## Immunofluorescence and image analysis

All immunostaining procedures were done as previously described (*Benner et al., 2023*). Antibodies and their respective dilutions are indicated in *Supplementary file 6*. Imaging was also completed as previously described (*Benner et al., 2023*).

## RNA-seq library preparation and sequencing

Twenty, 1-day-old post-eclosion $ovo^{\Delta BP}/ovo^{ovo-GAL4}$; *UASp-GFP* and $ovo^{\Delta BP}/ovo^{ovo-GAL4}$; *UASp-3xFHA-OVO-B* ovaries were dissected and germariums through previtellogenic egg chambers were removed with microdissection scissors and placed in ice cold PBS making up one biological replicate. RNA was then extracted from four biological replicates with a Qiagen RNeasy Plus Kit (QIAGEN) according to the manufacturer's protocol, eluted in dH$_2$O, and RNA concentrations were measured with Quant-iT RiboGreen RNA Assay Kit (Thermo Fisher Scientific). A total of 500 ng of total RNA was then used to make RNA-seq libraries with an Illumina Stranded mRNA Prep Kit according to the manufacturer's protocol (Illumina). IDT for Illumina RNA UD Indexes Set A were used. Library concentrations were measured with Quant-iT PicoGreen dsDNA Assay Kit (Thermo Fisher Scientific), pooled, and then 50 nucleotide paired-end DNA sequencing was completed on an Illumina NovaSeq 6000 system using a S1 flow cell (Illumina). Raw RNA-seq reads are available at the SRA (SRA26854132–26854139, 26854148–26854151).

## ChIP-seq library preparation and sequencing

Adult $ovo^{Cterm-3xFHA}$ and $ovo^{Cterm-GFP}$ females were collected and fed for 24 hr before ovaries were dissected. Fifty dissected ovaries were placed in ice cold phosphate buffered saline (PBS, Gibco, ThermoFisher Scientific) and then incubated in 1 mL crosslinking solution containing 2% formaldehyde (Pierce, ThermoFisher Scientific) (50 mM HEPES Buffer, 1 mM EDTA, 0.5 mM EGTA, 100 mM NaCl), and rotated at 37 °C for 20 min. Ovaries were then incubated in 1 mL stop solution (125 mM Glycine, 0.01% Triton X-100 (Millipore Sigma), diluted in PBS) and rotated for 5 min at room temperature. Ovaries were then washed twice with 1 mL ice cold wash buffer (0.01% Triton X-100 in PBS) for 5 min. The last wash was removed, and ovaries were stored at –80 °C until future processing. Once all samples were collected, 4x50 ovaries were then homogenized in 250 μL RIPA lysis buffer (Pierce, ThermoFisher Scientific) containing 1 x protease inhibitor cocktail (cOmplete Mini Protease Inhibitor Cocktail, Roche, Millipore Sigma) and 1 mM PMSF (Roche, Millipore Sigma) and kept on ice for 10 min. Forty mg of 212–300 μm acid-washed glass beads (Millipore Sigma) were then added to homogenized ovary lysate. Samples were then sonicated with a Bioruptor Pico sonication device (Diagenode) at 4 °C for 15 cycles of 30 s on and 30 s off. Sonicated lysate was then transferred to a new tube and centrifuged at 13,300 rpm for 10 min at 4 °C. Three supernatants were then combined to form one biological replicate. A total of 100 μL for each biological replicate was removed and stored at –80 °C for input control. To pull down C-terminally tagged OVO, 100 μL of monoclonal anti-HA-agarose (Millipore Sigma) or 50 μL of ChromoTek GFP-Trap agarose (Proteintech) were washed three times with RIPA lysis buffer and spun down at 1,200 RPMs for 1 min at 4 °C. 550 μL of $ovo^{Cterm-3xFHA}$ supernatant was added to monoclonal anti-HA-agarose and 550 μL of $ovo^{Cterm-GFP}$ supernatant was added to ChromoTek GFP-Trap agarose. Samples were supplemented with 1 x protease inhibitor cocktail and 1 mM PMSF and incubated on a rotator at 4 °C overnight.

The next day, agarose was washed in a stepwise fashion with solutions from a Chromatin Immunoprecipitation Assay Kit (Millipore Sigma), beginning with 1 mL of a low salt wash buffer, high salt wash buffer, LiCl buffer, and ending with 2 washes in 0.1 x TE buffer. A total of 300 μL of freshly prepared ChIP elution/decrosslinking solution (1% SDS, 100 mM NaHCO$_3$, 250 mM NaCl, 10 mM EDTA, 50 mM Tris-HCl, 200 μg/mL Proteinase K) was added to the pelleted agarose, or 200 μL of chip elution/decrosslinking solution was added to 100 μL input control and incubated at 65 °C overnight. DNA was extracted by adding 300 μL phenol:chloroform:iso-amyl alcohol (125:24:1) (Millipore Sigma). The samples were vortexed for 30 s then centrifuged at 13,300 RPMs for 5 min at 4 °C. The aqueous layer was extracted, and this process was repeated once more. One μL glycogen (20 mg/mL), 30 μL 1 M sodium acetate, and 750 μL 100% EtOH was added to the extracted aqueous layer, vortexed, and incubated at –20 °C for 30 min. Solution was spun at 13,300 RPMs for 20 min at 4 °C. Supernatant was removed and the pellet was washed with 500 μL 70% EtOH and spun down at 13,300 RPMs for 20 min

at 4 °C. This step was repeated but with 100% EtOH. The resulting pellet was briefly speedvacced and resuspended in 50 µL dH$_2$O.

To make ChIP-seq libraries, DNA concentration for immunoprecipitated and input control samples were measured with a Quant-iT PicoGreen dsDNA Assay Kit (ThermoFisher Scientific). Five ng of DNA for each sample was then used with the NEBNext Ultra II DNA Library Prep Kit for Illumina (New England Biolabs) and completed according to the manufacturer's protocol. ChIP-seq library concentrations were then measured with a Quant-iT PicoGreen dsDNA Assay Kit, pooled, and then 50 nucleotide paired-end DNA sequencing was performed on an Illumina NovaSeq 6000 system using the XP workflow (Illumina). Raw ChIP-seq reads are available at the SRA (SRA26854140–26854147).

## RNA-seq, ChIP-seq, CAGE-seq and gene ontology analysis

For RNA-seq analysis of *ovo$^{ΔBP}$/ovo$^{ovo-GAL4}$*; *UASp-GFP* and *ovo$^{ΔBP}$/ovo$^{ovo-GAL4}$*; *UASp-3xFHA-OVO-B* ovaries, 50 nucleotide paired-end reads were mapped to the FlyBase r6.46 genome (*Gramates et al., 2022*) for differential expression analysis and the BDGP Release 6 *Drosophila* Genome (*dos Santos et al., 2015*) for read level genome browser tracks using Hisat2 (-k 1 `--rna-strandness` RF `--dta`) (*Kim et al., 2019*). DNA sequences for *GAL4* and *GFP* were added to the FlyBase r6.46 genome as separate chromosomes. Mapped reads were then sorted and indexed with Samtools (samtools sort and samtools index) (*Danecek et al., 2021*). Gene level readcounts were then derived with htseq-count (-s reverse -r pos) (*Anders et al., 2015*) and used for differential expression analysis with DeSeq2 (*Love et al., 2014*). Genes with 0 mapped reads were removed from the DESeq2 analysis.

For ChIP-seq analysis of OVO-HA, OVO-GFP, OVO-HA input, and OVO-GFP input samples, 50 nucleotide paired-end reads were mapped to the FlyBase r6.46 genome for peak calling analysis and the BDGP Release 6 *Drosophila* Genome for read level genome browser tracks using Hisat2 (-k 1 `--no-spliced-alignment` -X 900). Mapped reads were sorted using Samtools (samtools sort and samtools index) and duplicate reads were removed with Picard (REMOVE_DUPLICATES = true) (*broadinstitute, 2024*). Significant ChIP peaks were called for OVO-HA and OVO-GFP versus their respective input controls separately using Macs3 callpeak software (-g 1.2e8 -q 0.0001) (*Zhang et al., 2008*). Overlapping ChIP peaks for OVO-HA and OVO-GFP were then determined with bedtools intersect software (*Quinlan and Hall, 2010*). Peak calling for GSC ATAC-seq (SRR24203655), 32 c ATAC-seq (SRR24203650), GSC H3K27ac (SRR11084657), H3K4me3 (SRR11084658), H3K27me3 (SRR11084656), H3K9me3 (SRR24203631), 8 c NC H3K4me3 (SRR24203629), 32 c NC H3K27ac (SRR24203635), and H3K27me3 (SRR11084652) ChIP-seq versus their respective input controls (SRR11084655, SRR11084651, SRR24203634, SRR24203637) was conducted in the same manner as OVO ChIP-seq.

In order to generate gene-level read coverage tracks, deepTools' bamCompare software was used to generate a single bigWig file comparing all replicates versus input controls (-bs 5 `--effectiveGenomeSize 142573017 --normalizeUsing BPM --exactScaling --scaleFactorsMethod None`) (*Ramírez et al., 2016*). The bigWig file was then uploaded to UCSC genome browser for visualization (*Kent et al., 2002*).

To generate read coverage plots centered on the motif location or OVO peak maximums, genomic locations of significant scoring motifs or peak maximums within overlapping OVO ChIP peaks were determined and used as input for deepTools' computeMatrix reference-point (`-a 2000 -b 2000 -bs 25 --missingDataAsZero`). Read density profiles for each motif or OVO peak maximum were then visualized with deeptools plotProfile. In order to generate read coverage plots centered on the TSS, the same methods as above were conducted except genes overlapping OVO ChIP peaks containing the respective significant OVO DNA binding motifs were used as input instead.

In order to generate gene level read coverage heatmaps, deepTools' computeMatrix scale-regions software was used to generate a single matrix for genes that were bound by OVO (-bs 25 `--missingDataAsZero -m 4000 --metagene`). This matrix was then used as input for deepTools' plotHeatmap software to generate heatmaps of ChIP-seq and RNA-seq for the given OVO binding profiles centered on the TSS (--sortUsing max).

For CAGE-seq analysis, CAGE-seq libraries for ovary (SRR488283, SRR488282), testes (SRR488284, SRR488308, SRR488285, SRR488309), and male and female digestive system (SRR488289, SRR488288) tissues were downloaded and combined for each tissue type from the SRA. Reads were mapped to the BDGP Release 6 *Drosophila* Genome with Hisat2 (-k 1). Mapped reads were sorted with Samtools.

Significant dominant TSSs were then determined with CAGEr software (*Haberle et al., 2015*) from sorted BAM files with getCTSS and annotated with annotateCTSS using the dm6.ensGene.gtf file (*Hubbard et al., 2002*) downloaded from UCSC (*Kent et al., 2002*). CAGE-seq reads were normalized with normalizeTagCount (ce, method = "simpleTpm", fitInRange = c(5, 40000), alpha = 1.15, T=1*10^6) and then TSS clusters were determined with clusterCTSS (ce, threshold = 1, thresholdIsTpm = TRUE, nrPassThreshold = 1, method = "paraclu", maxDist = 20, removeSingletons = TRUE, keepSingletonsAbove = 5) in order to determine the dominant significant TSS for each respective tissue.

Gene ontology enrichment analysis was completed with g:Profiler's g:GOSt software (*Raudvere et al., 2019*) on the set of genes overlapping OVO ChIP peaks over the TSS and significantly upregulated in the presence of ectopic OVO (525 genes in total). All genes that were considered to be expressed in our RNA-seq datasets were used as a background control (10,801 genes in total). Default parameters were used for the enrichment analysis except for 'statistical domain scope' was set to 'custom' (our control background genes were uploaded here), 'significance threshold' was set to 'Bonferroni correction', and only GO biological process terms were searched for enrichment with the gene list. The GO terms listed in *Figure 6* represent the 24 smallest GO term sizes according to *Supplementary file 5*.

Fisher's exact test was conducted for each respective analysis with the fisher.test() command in R (*R Development Core Team, 2021*).

## de novo Motif enrichment and promoter motif analysis

DNA sequences from significant overlapping OVO ChIP peaks were extracted from the *Drosophila* r6.46 genome and submitted to STREME software (*Bailey, 2021*) of the MEME suite (*Bailey et al., 2015*). The default parameters were used for de novo motif enrichment analysis, including the use of shuffled input sequences as a control. After identifying 'OVO Motif One', OVO ChIP peaks that contained that sequence were removed and the resulting ChIP peaks were resubmitted for STREME analysis deriving derivative OVO DNA binding motifs like above. Significant OVO DNA binding motifs and in vitro OVO DNA binding motifs were searched in the BDGP Release 6 *Drosophila* Genome using FIMO (*Grant et al., 2011*). In order to find significant DNA binding motif matches for 'OVO Motif One', this motif from STREME was submitted to Tomtom software (*Gupta et al., 2007*) of the MEME suite and searched within the JASPAR Core Insect database (2022) (*Castro-Mondragon et al., 2022*). XSTREME software was used with the default parameters to identify previously characterized and published DNA binding motifs within OVO ChIP peaks (*Grant and Bailey, 2021*).

Promoter motif analysis was conducted by extracting the DNA sequences 200 nucleotides upstream and downstream of the significant dominant TSSs from CAGE-seq analysis for each respective tissue type. All common core promoter motifs (*FitzGerald et al., 2006*; *Ohler et al., 2002*) were then searched in these sequences depending on their strand specificity with the use of FIMO from the MEME suite using a p-value of <0.003 for all non-OVO promoter motifs. All OVO motifs found in this study and through in vitro methods were also searched with the same method, except a p-value of <0.0002 was used.

## Acknowledgements

We thank previous and current members of the Oliver lab, Laboratory of Biochemistry and Genetics at NIH, and L.B. committee members Mark Van Doren and Allan Spradling for insightful discussion and comments on this work throughout. Monoclonal antibodies were obtained from the Developmental Studies Hybridoma Bank, created by the Eunice Kennedy Shriver National Institute of Child Health and Human Development (NICHD) of the NIH and maintained at the Department of Biology, University of Iowa, Iowa City, IA 52242. Genetic and genomic information was obtained from FlyBase (U41 HG-000739). This work utilized the computational resources of the NIH High-Performance Computing Biowulf cluster (http://hpc.nih.gov). Sequencing was completed by The National Heart, Lung, and Blood Institute (NHLBI) DNA Sequencing and Genomics Core. Funding this research was supported in part by the Intramural Research Program of the NIH, The National Institute of Diabetes and Digestive and Kidney Diseases (NIDDK) (awarded to BO). LB was supported by the NIH Graduate Partnership Program.

# Additional information

## Funding

| Funder | Grant reference number | Author |
|---|---|---|
| National Institute of Diabetes and Digestive and Kidney Diseases | Intramural Research Program | Brian Oliver |

The funders had no role in study design, data collection and interpretation, or the decision to submit the work for publication.

## Author contributions

Leif Benner, Conceptualization, Data curation, Formal analysis, Investigation, Visualization, Methodology, Writing – original draft, Writing – review and editing; Savannah Muron, Jillian G Gomez, Data curation, Formal analysis, Investigation, Visualization, Methodology, Writing – review and editing; Brian Oliver, Conceptualization, Supervision, Funding acquisition, Methodology, Project administration, Writing – review and editing

## Author ORCIDs

Leif Benner ⓘ https://orcid.org/0000-0002-3716-4522
Brian Oliver ⓘ https://orcid.org/0000-0002-3455-4891

Reviewer #1 (Public Review): https://doi.org/10.7554/eLife.94631.3.sa1
Author response https://doi.org/10.7554/eLife.94631.3.sa2

# Additional files

## Supplementary files

- Supplementary file 1. OVO ChIP-seq results by chromosome.
- Supplementary file 2. Genomic locations of significant overlapping OVO ChIP peaks.
- Supplementary file 3. Significant OVO DNA binding motifs in MEME format.
- Supplementary file 4. Differential expression analysis results for all genes.
- Supplementary file 5. Significantly enriched GO biological process terms.
- Supplementary file 6. Flybase ART table.
- MDAR checklist

## Data availability

Sequencing data has been deposited to the SRA under accession numbers SRR26854132– SRR26854151 (BioProject: PRJNA1041436).

The following dataset was generated:

| Author(s) | Year | Dataset title | Dataset URL | Database and Identifier |
|---|---|---|---|---|
| Benner L | 2023 | ChIP of OVO-GFP and OVO-HA in *Drosophila* adult ovaries. RNA-seq of ovo hypomorphic and rescue alleles | https://www.ncbi.nlm.nih.gov/bioproject/?term=PRJNA1041436 | NCBI BioProject, PRJNA1041436 |

The following previously published datasets were used:

| Author(s) | Year | Dataset title | Dataset URL | Database and Identifier |
|---|---|---|---|---|
| Chen ZX, Sturgill D, Qu J, Jiang H, Park S, Boley N, Suzuki AM, Fletcher AR, Plachetzki DC, FitzGerald PC, Artieri CG, Atallah J, Barmina O, Brown JB, Blankenburg KP, Clough E, Dasgupta A, Gubbala S, Han Y, Jayaseelan JC, Kalra D, Kim YA, Kovar CL, Lee SL, Li M, Malley JD, Malone JH, Mathew T, Mattiuzzo NR, Munidasa M, Muzny DM, Ongeri F, Perales L, Przytycka TM, Pu LL, Robinson G, Thornton RL, Saada N, Scherer SE, Smith HE, Vinson C, Warner CB, Worley KC, Wu YQ, Zou X, Cherbas P, Kellis M, Eisen MB, Piano F, Kionte K, Fitch DH, Sternberg PW, Cutter AD, Duff MO, Hoskins RA, Graveley BR, Gibbs RA, Bickel PJ, Kopp A, Carninci P, Celniker SE, Oliver B, Richards S | 2014 | Comparative validation of the *D. melanogaster* modENCODE transcriptome annotation | https://www.ncbi.nlm.nih.gov/bioproject/PRJNA75285 | NCBI BioProject, PRJNA75285 |
| DeLuca SZ, Ghildiyal M, Pang LY, Spradling AC | 2020 | Differentiating *Drosophila* female germ cells initiate Polycomb silencing by regulating PRC2-interacting proteins | https://www.ncbi.nlm.nih.gov/bioproject/PRJNA606593 | NCBI BioProject, PRJNA606593 |
| Pang LY, DeLuca S, Zhu H, Urban JM, Spradling AC | 2023 | Chromatin and gene expression changes during female *Drosophila* germline stem cell development illuminate the biology of highly potent stem cells | https://www.ncbi.nlm.nih.gov/bioproject/?term=PRJNA956868 | NCBI BioProject, PRJNA956868 |

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
