## [Editor Report · eLife assessment]

This **useful** manuscript extends prior work to identify OVO as a major transcriptional activator of the female germline gene expression program. Using a combination of **solid** genomic strategies, the authors demonstrate that OVO binds to the promoters of hundreds of genes in the female germline and promotes their expression.

---

## [Referee Report · Reviewer #1 (Public Review)]

Summary:

In this manuscript, Benner et al. identify OVO as a transcriptional factor instrumental in promoting expression of hundreds of genes essential for female germline identity and early embryo development. Prior data had identified both ovo and otu as genes activated by OVO binding to the promoters. By combining ChIP-seq, RNA-seq and analysis of prior datasets, the authors extend these data to hundreds of genes and therefore propose that OVO is a master transcriptional regulator of oocyte development. They further speculate that OVO may function to promote chromatin accessibility to facilitate germline gene expression. Overall, the data compellingly demonstrate a much broader role for OVO in activation of genes in the female germline than previously recognized. By contrast, the relationship between OVO, chromatin accessibility and the timing of gene expression is only correlative, and more work will be needed to determine the mechanisms by which OVO promotes transcription.

Strengths

Here Benner at al. convincingly show that OVO is a transcriptional activator that promotes expression of hundreds of genes in the female germline. The ChIP-seq and RNA-seq data included in the manuscript are robust and the analysis is compelling.

Importantly, the set of genes identified are essential for maternal processes, including egg production and patterning of the early embryo. Together, these data identify OVO as a major transcriptional activator of the numerous genes expressed in the female germline, deposited into the oocyte and required for early gene expression. This is an important finding as this is an essential process for development and prior to this study the major drivers of this gene expression program were unknown.

Weaknesses

The novelty of the manuscript is somewhat limited as the authors show that, like two prior, well-studied OVO target genes, OVO binds to promoters of germline genes and activates transcription. The fact that OVO performs this function more broadly is not particularly surprising.

A major challenge to understanding the impact of this manuscript is the fact that the experimental system for the RNA-seq, the tagged constructs, and the expression analysis that provides the rationale for the proposed pioneering function of OVO are all included in a separate manuscript.

---

## [Author Response]

The following is the authors’ response to the original reviews.

**Public Reviews:**

**Reviewer #1 (Public Review):**
Summary:In this manuscript, Benner et al. identify OVO as a transcriptional factor instrumental in promoting the expression of hundreds of genes essential for female germline identity and early embryo development. Prior data had identified both ovo and otu as genes activated by OVO binding to the promoters. By combining ChIP-seq, RNA-seq, and analysis of prior datasets, the authors extend these data to hundreds of genes and therefore propose that OVO is a master transcriptional regulator of oocyte development. They further speculate that OVO may function to promote chromatin accessibility to facilitate germline gene expression. Overall, the data compellingly demonstrate a much broader role for OVO in the activation of genes in the female germline than previously recognized. By contrast, the relationship between OVO, chromatin accessibility, and the timing of gene expression is only correlative, and more work will be needed to determine the mechanisms by which OVO promotes transcription.

We fully agree with this summary.

Strengths:Here Benner et al. convincingly show that OVO is a transcriptional activator that promotes expression of hundreds of genes in the female germline. The ChIP-seq and RNA-seq data included in the manuscript are robust and the analysis is compelling.Importantly, the set of genes identified is essential for maternal processes, including egg production and patterning of the early embryo. Together, these data identify OVO as a major transcriptional activator of the numerous genes expressed in the female germline, deposited into the oocyte and required for early gene expression. This is an important finding as this is an essential process for development and prior to this study, the major drivers of this gene expression program were unknown.

We are delighted that this aspect of the work came across clearly. Understanding the regulation of maternal effect genes has been something of a black-box, despite the importance of this class of genes in the history of developmental genetics. The repertoire of essential oogenesis/embryonic development genes that are bound by and respond to OVO are well characterized in the literature, but nothing is known about how they are transcriptionally regulated. We feel the manuscript will be of great interest to readers working on these genes.

Weaknesses:The novelty of the manuscript is somewhat limited as the authors show that, like two prior, well-studied OVO target genes, OVO binds to promoters of germline genes and activates transcription. The fact that OVO performs this function more broadly is not particularly surprising.

Clearly, transcription factors regulate more than one or two genes. Never-the-less we were surprised at how many of the aspects of oogenesis *per se* and maternal effect genes were OVO targets. It was our hypothesis that OVO would have a transcriptional effect genome-wide, however, it was less clear whether OVO would always bind at the core promoter, as is with the case of *ovo* and *otu*. Our results strongly support the idea that core promoter proximal binding is essential for OVO function; a conclusion of work done decades ago, which has not been revisited using modern techniques.

A major challenge to understanding the impact of this manuscript is the fact that the experimental system for the RNA-seq, the tagged constructs, and the expression analysis that provides the rationale for the proposed pioneering function of OVO are all included in a separate manuscript.

This is a case where we ended up with a very, very long manuscript which included a lot of revisiting of legacy data. It was a tough decision on how to break up all the work we had completed on *ovo* to date. In our opinion, it was too much to put everything into a single manuscript unless we wanted a manuscript length supplement (we were also worried that supplemental data is often overlooked and sometimes poorly reviewed). We therefore decided to split the work into a developmental localization/characterization paper and a functional genomics paper. As it stands both papers are long. Certainly, readers of this manuscript will benefit from reading our previous OVO paper, which we submitted before this one. The earlier manuscript is under revision at another journal and we hope that this improved manuscript will be published and accessible shortly.

**Reviewer #2 (Public Review):**
Summary:In this manuscript, Benner et al. interrogate the transcriptional regulator OVO to identify its targets in the *Drosophila* germline. The authors perform ChIP-seq in the adult ovary and identify established as well as novel OVO binding motifs in potential transcriptional targets of OVO. Through additional bioinformatic analysis of existing ATAC-seq, CAGE-seq, and histone methylation data, the authors confirm previous reports that OVO is enriched at transcription start sites and suggest that OVO does not act as part of the core RNA polymerase complex. Benner et al. then perform bulk RNA-seq in OVO mutant and "wildtype" (GAL4 mediated expression of OVO under the control of the ovo promoter in OVO mutants) ovaries to identify genes that are differentially expressed in the presence of OVO. This analysis supports previous reports that OVO likely acts at transcription start sites as a transcriptional activator. While the authors propose that OVO activates the expression of genes that are important for egg integrity, maturation, and for embryonic development (nanos, gcl, pgc, bicoid), this hypothesis is based on correlation and is not supported by in vivo analysis of the respective OVO binding sites in some of the key genes. A temporal resolution for OVO's role during germline development and egg chamber maturation in the ovary is also missing. Together, this manuscript contains relevant ChIP-seq and RNA-seq datasets of OVO targets in the *Drosophila* ovary alongside thorough bioinformatic analysis but lacks important in vivo experimental evidence that would validate the high-quality datasets.

We thank reviewer 2 for the appreciation of the genomics data and analysis. Some of the suggested in vivo experiments are clear next steps, which are well underway. These are beyond the scope of the current manuscript.

Temporal analysis of *ovo* function in egg chamber development is not easy, as only the weakest *ovo* alleles have any egg chambers to examine. However, we will also point out the long-known phenotypes of some of those weak alleles in the text (e.g. ventralized chambers in ovoD3/+). We will need better tools for precise rescue/degradation during egg chamber maturation.

Strengths:The manuscript contains relevant ChIP-seq and RNA-seq datasets of OVO targets in the *Drosophila* ovary alongside thorough bioinformatic analysis

Thank you. We went to great lengths to do our highly replicated experiments in multiple ways (e.g. independent pull-down tags) and spent considerable time coming up with an optimized and robust informatic analysis.

Weaknesses:(1) The authors propose that OVO acts as a positive regulator of essential germline genes, such as those necessary for egg integrity/maturation and embryonic/germline development. Much of this hypothesis is based on GO term analysis (and supported by the authors' ChIP-seq data). However accurate interpretation of GO term enrichment is highly dependent on using the correct background gene set. What control gene set did the authors use to perform GO term analysis (the information was not in the materials and methods)? If a background gene set was not previously specified, it is essential to perform the analysis with the appropriate background gene set. For this analysis, the total set of genes that were identified in the authors' RNA-seq of OVO-positive ovaries would be an ideal control gene set for which to perform GO term analysis. Alternatively, the total set of genes identified in previous scRNA-seq analysis of ovaries (see Rust et al., 2020, Slaidina et al., 2021 among others) would also be an appropriate control gene set for which to perform GO term analysis. If indeed GO term analysis of the genes bound by OVO compared to all genes expressed in the ovary still produces an enrichment of genes essential for embryonic development and egg integrity, then this hypothesis can be considered.

We feel that this work on OVO as a positive regulator of genes like *bcd*, *osk*, *nos*, *png*, *gnu*, *plu*, etc., is closer to a demonstration than a proposition. These are textbook examples of genes required for egg and early embryonic development. Hopefully, this is not lost on the readers by an over-reliance on GO term analysis, which is required but not always useful in genome-wide studies.

We used GO term enrichment analysis as a tool to help focus the story on some major pathways that OVO is regulating. To the specific criticism of the reference gene-set, GO term enrichment analysis in this work is robust to gene background set. We will update the GO term enrichment analysis text to indicate this fact and add a table using expressed genes in our RNA-seq dataset to the manuscript and clarify gene set robustness in greater detail in the methods of the revision. We will also try to focus the reader’s attention on the actual target genes rather than the GO terms in the revised text.

We have updated the GO term analysis by including all the expressed genes in our RNA-seq datasets as a background control. Figure 6 has been updated to include the significant GO terms. We have outlined changes in the methods section below.

Lines 794-801:

“Gene ontology enrichment analysis was completed with g:Profiler’s g:GOSt software (Raudvere et al. 2019) on the set of genes overlapping OVO ChIP peaks over the TSS and significantly upregulated in the presence of ectopic OVO (525 genes in total). All genes that were considered to be expressed in our RNA-seq datasets were used as a background control (10,801 genes in total). Default parameters were used for the enrichment analysis except for ‘statistical domain scope’ was set to ‘custom’ (our control background genes were uploaded here), ‘significance threshold’ was set to ‘Bonferroni correction’, and only GO biological process terms were searched for enrichment with the gene list. The GO terms listed in Figure 6 represent the 24 smallest GO term sizes according to Table S5.”

(2) The authors provide important bioinformatic analysis of new and existing datasets that suggest OVO binds to specific motifs in the promoter regions of certain germline genes. While the bioinformatic analysis of these data is thorough and appropriate, the authors do not perform any in vivo validation of these datasets to support their hypotheses. The authors should choose a few important potential OVO targets based on their analysis, such as gcl, nanos, or bicoid (as these genes have well-studied phenotypes in embryogenesis), and perform functional analysis of the OVO binding site in their promoter regions. This may include creating CRISPR lines that do not contain the OVO binding site in the target gene promoter, or reporter lines with and without the OVO binding site, to test if OVO binding is essential for the transcription/function of the candidate genes.

Exploring mechanism using in vivo phenotypic assays is awesome, so this is a very good suggestion. But, it is not essential for this work -- as has been pointed out in the reviews, in vivo validation of OVO binding sites has been comprehensively done for two target genes, *ovo* and *otu*. The “rules” appear similar for both genes. That said, we are already following up specific OVO target genes and the detailed mechanism of OVO function at the core promoter. We removed some of our preliminary in vivo figures from the already long current manuscript. We continue to work on OVO and expect to include this type of analysis in a new manuscript.

(3) The authors perform de novo motif analysis to identify novel OVO binding motifs in their ChIP-seq dataset. Motif analysis can be significantly strengthened by comparing DNA sequences within peaks, to sequences that are just outside of peak regions, thereby generating motifs that are specific to peak regions compared to other regions of the promoter/genome. For example, taking the 200 nt sequence on either side of an OVO peak could be used as a negative control sequence set. What control sequence set did the authors use as for their de novo motif analysis? More detail on this is necessary in the materials and methods section. Re-analysis with an appropriate negative control sequence set is suggested if not previously performed.

We apologize for being unclear on negative sequence controls in the methods. We used shuffled OVO ChIP-seq peak sequences as the background for the de novo motif analysis, which we will better outline in the methods of the revision. This is a superior background set of sequences as it exactly balances GC content in the query and background sequences. We are not fond of the idea of using adjacent DNA that won’t be controlled for GC content and shadow motifs. Furthermore, the de novo OVO DNA binding motifs are clear, statistically significant variants of the characterized in vitro OVO DNA binding motifs previously identified (Lu et al., 1998; Lee and Garfinkel, 2000; Bielinska et al., 2005), which lends considerable confidence. We also show that the OVO ChIP-seq read density are highly enriched for all our identified motifs, as well as the in vitro motifs. We provide multiple lines of evidence, through multiple methods, that the core OVO DNA binding motif is 5’-TAACNGT-3’. We have high confidence in the motif data.

We have added the below text to the methods section for further clarity on motif analysis parameters.

Lines 808-812

“The default parameters were used for de novo motif enrichment analysis, including the use of shuffled input sequences as a control. After identifying ‘OVO Motif One’, OVO ChIP peaks that contained that sequence were removed and the resulting ChIP peaks were resubmitted for STREME analysis deriving derivative OVO DNA binding motifs like above.”

(4) The authors mention that OVO binding (based on their ChIP-seq data) is highly associated with increased gene expression (lines 433-434). How many of the 3,094 peaks (conservative OVO binding sites), and what percentage of those peaks, are associated with a significant increase in gene expression from the RNA-seq data? How many are associated with a decrease in gene expression? This information should be added to the results section.

Not including the numbers of the overlapping ChIP peaks and expression changes in the text was an oversight on our part. The numbers that relate to this (666 peaks overlapping genes that significantly increased in expression, significant enrichment according to Fishers exact test, 564 peaks overlapping genes that significantly decreased in expression, significant depletion according to Fishers exact test) are found in figure 4C and will be added to the text.

We have modified the results section to include the overlap between the RNA-seq and ChIP-seq data.

Lines 463-468

“We found that 2,298 genes that were expressed in our RNA-seq data overlapped an OVO ChIP peak. 666 genes significantly increased in expression and were bound by OVO, which is a significant enrichment according to a Fisher’s exact test (Figure 4C, cyan dots, p < 0.01, odds ratio = 2.21). While conversely, 564 genes decreased in expression and were bound by OVO, indicating a significant depletion according to a Fisher’s exact test (Figure 4C, blue dots, p < 0.01, odds ratio = 0.85).”

(5) The authors mention that a change in endogenous OVO expression cannot be determined from the RNA-seq data due to the expression of the OVO-B cDNA rescue construct. Can the authors see a change in endogenous OVO expression based on the presence/absence of OVO introns in their RNA-seq dataset? While intronic sequences are relatively rare in RNA-seq, even a 0.1% capture rate of intronic sequence is likely to be enough to determine the change in endogenous OVO expression in the rescue construct compared to the OVO null.

This is a good point. The *GAL4* transcript is downstream of *ovo* expression in the hypomorphic *ovoovo-GAL4* allele. We state in the text that there is a nonsignificant increase in *GAL4* expression with ectopic rescue OVO, although the trend is positive. We calculated the RPKM of RNA-seq reads mapping to the intron spanning exon 3 and exon 4 in *ovo-RA* and found that there is also a nonsignificant increase in intronic RPKM with ectopic rescue OVO (we will add to the results in the revision). We would expect OVO to be autoregulatory and potentially increase the expression of *GAL4* and/or intronic reads, but the *ovoovoGAL4*>*UASp-OVOB* is not directly autoregulatory like the endogenous locus. It is not clear to us how the intervening GAL4 activity would affect OVOB activity in the artificial circuit. Dampening? Feed-forward? Is there an effect on OVOA activity? Regardless, this result does not change our interpretation of the other OVO target genes.

We have added the analysis of intronic *ovo* RNA-seq to the results as outlined below.

Lines 512-520

“Transcriptionally, *ovo* RNA-seq reads are likely derived from the *UASp-3xFHA-OVO-B* cDNA rescue or are indistinguishable between the genomic locus and rescuing cDNA transgene. We found a nonsignificant increase in exon 3 to exon 4 intronic *ovo* reads with the expression of ectopic rescue OVO (log2 fold change = 0.76, p-adj = 0.26). These intronic reads would be derived from the endogenous *ovo* locus, but it is difficult to conclusively determine if the endogenous *ovo* locus would respond transcriptionally to ectopic OVO downstream of *UASp* (for example, the pathway for *ovo* is no longer autoregulatory in *ovoovo-GAL4*/*ovoΔBP*; *UASp-3xFHA-OVO-B* germ cells, there is an additional GAL4>*UASp* activation step). So, we could not confidently assess whether *ovo* responded transcriptionally to ectopic rescue OVO.”

(6) The authors conclude with a model of how OVO may participate in the activation of transcription in embryonic pole cells. However, the authors did not carry out any experiments with pole cells that would support/test such a model. It may be more useful to end with a model that describes OVO's role in oogenesis, which is the experimental focus of the manuscript.

We did not complete any experiments in embryonic pole cells in this manuscript and base our discussion on the potential dynamics of OVO transcriptional control and our previous work showing maternal and zygotic OVO protein localization in the developing embryonic germline. Obviously, we are highly interested in this question and continue to work on the role of maternal OVO. We agree that we are extended too far and will remove the embryonic germ cell model in the figure. We will instead focus on the possible mechanisms of OVO gene regulation in light of the evidence we have shown in the adult ovary, as suggested.

We have removed figure 7 and have re-written the last two paragraphs of the discussion as below.

Lines 645-663

“The requirement for OVO at the TSS of target genes has been well characterized at its own locus as well as its downstream target *otu*. Our OVO ChIP and expression data confirm findings from previous work that OVO is binding to these target promoters, and in the case of *otu*, strongly responds transcriptionally to the presence of OVO. Although we did not test the requirement for OVO DNA binding motifs at other OVO bound genes in this work, this has been extensively explored before, showing that removal of OVO

DNA binding sites overlapping the TSS results in a strong decrease in reporter expression (Lü et al. 1998; Bielinska et al. 2005; Lü and Oliver 2001). Removal of more distal upstream OVO DNA binding sites also reduces reporter expression to a lesser degree. However, for most cases tested, removal of OVO DNA binding sites while leaving the rest of the enhancer regions intact, never totally abolished reporter expression. These dynamics are highly similar to work that has been completed on the pioneer factor *zelda* (*zld*). Adding *zld* DNA binding motifs to a stochastically expressed transcriptional reporter increases the activity and response of the reporter (Dufourt et al. 2018). Distally located *zld* DNA binding motifs influenced reporter expression to a lesser degree than proximal sites. A single *zld* DNA binding site adjacent to the TSS produced the strongest reporter activity. Importantly, just like the activity of OVO transgenic reporters, there is not an absolute requirement for *zld* DNA binding to activate reporter expression, however, the addition of TSS adjacent *zld* DNA binding motifs does strongly influence reporter response. We know that *zld* achieves this reporter response through its pioneering activity (Xu et al. 2014; Harrison et al. 2011), whether OVO achieves this similar effect on gene expression through a shared mechanism, or in cooperation with other transcription factors needs to be further explored.”

**Recommendations for the authors:**

**Reviewer #1 (Recommendations For The Authors):**
The Results section could be streamlined by limiting the discussion of analysis to only those details that are unusual or essential for understanding the science. For example, the fact that MACS3 was used to call peaks seems most suitable for the Methods section.

We have removed the below excerpts from the results section to streamline the text.

‘We compared immuno-purified OVO associated DNA with input DNA as a control, for a total of 12 ChIPseq libraries, which we sequenced using the Illumina system. After quality control and alignment to the *Drosophila* r6.46 genome (Gramates et al. 2022), we used MACS3 (Zhang et al. 2008)’

The Supplemental Tables are referred to out of order. Table S2 is referred to on line 143 while Table S1 is not referred to until the Methods section.

We have reorganized the order of the tables in the manuscript text.

In the analysis of CAGE-seq data, it is unclear whether there is anything distinctive about the ~2000 regions bound by OVO but that is not near TSS in the ovary dataset. Are these TSS that are not active in the ovary or are these non-promoter bound OVO sites? If they are TSS of genes not in the CAGE-seq data set, are these genes expressed in other tissues or just expressed at lower levels in the ovary?

This was a good point that prompted us to take a closer look at the characteristics of OVO binding and its relationships to promoters and other gene elements. 45% of OVO ChIP peaks overlapped the TSS while 55% were either non-overlapping downstream or upstream of the TSS. When plotting OVO ChIP read density, there was still a striking enrichment of OVO binding over the TSS, even though the ChIP peak was not overlapping the TSS (new figure 1K). This is possibly due to weaker direct OVO binding at the TSS that was not considered significant in the peak calling software or were indirect interactions of the distal OVO binding and the TSS. We outline this in the below text added to the results section on the OVO ChIP. To showcase these results, we have included a new panel in figure 1K. We removed the panel showing the enrichment over the cage-seq TSS, but this same data remains in the heatmap shown in figure 1L, so no information is lost. To directly answer the Cage-seq questions considering the OVO bound over the annotated TSS results, we found that 1,047 chip peaks overlapped CAGE-seq TSS, which is only 347 fewer than the annotated TSS overlap (1,394). Of the 1,394 genes that were bound by over the TSS, all of them were considered to be expressed in our RNA-seq dataset, indicating that these might just be more lowly expressed genes that for whatever reason were not considered to be enriched TSSs in the CAGE-seq data. This difference is likely not significant.

Lines 235-251

“Although OVO ChIP peaks overlapping genes showed a strong read density enrichment over the TSS, we found that only 45% (1,394/3,094) of OVO ChIP peaks directly overlapped a TSS. 43% (1,339/3,094) of OVO ChIP peaks were found to overlap the gene body downstream of the TSS (intronic and exonic sequences) and 12% (366/3,094) did not overlap any gene elements, indicating that they were intergenic.

We were interested in the differences between OVO binding directly over the TSS or at more distal upstream and downstream sites. We decided to plot the OVO ChIP read density of these different classes of OVO binding patterns and found that OVO bound over the TSS produced a sharp read density enrichment over the TSS which was consistent with what was found for all OVO bound genes (Figure 1K). OVO binding along the gene body surprisingly also showed a read density enrichment over the TSS, although the magnitude of read density enrichment was notably less than TSS OVO binding. Intergenic OVO binding also showed these same characteristics with a notable upstream read density enrichment possibly indicative of enhancer binding. This indicates that although the significantly called OVO ChIP peaks did not overlap the TSS, there was still a propensity for TSS sequences to be enriched with OVO ChIP over the input control. This could be due to weaker direct in vivo binding of OVO to these TSSs or indirect interactions between the upstream/downstream OVO bound sequences and the TSS, possibly through a looping enhancer-promoter interaction. However, regardless of the location of the OVO ChIP peak, OVO seemed to always be enriched at or in close proximity to TSSs.”

It would be helpful for the authors to provide a bit more detailed analysis of chromatin states of OVObound regions in GSC, 8c NC, and 32c NC (or some more clarity in the current analysis). Are the regions that are bound by OVO accessible in all these cell types or specifically enriched for accessibility in a subset? The authors state that OVO binding is correlated with open chromatin, but whether these are regions that are open in all cell types analyzed or a subset is not clear from the data presented. Promoters are often accessible regardless of cell type, so it is unclear what exactly is to be concluded from this association. Also, is the proximity to open chromatin features for OVO-bound promoters (as shown in Figure 2C) different than non-OVO-bound promoters (the two classes shown Figure 1L, for example)?

We utilized previously published datasets of staged germ cell chromatin status to look at the association of chromatin status and OVO binding. Unfortunately, not all the same germ cell stages were profiled for each chromatin mark from the datasets derived for these two papers. For example, only H3K4me3 data exists for GSCs, and only gsc and 8c data exists for H3K9me3, while the other chromatin marks had more profiles, even including later stages. We focused specifically on gsc and 32c (essentially stage 5 egg chambers) for the other chromatin marks since that is when the *ovo* hypomorphic egg chambers arrest. A nice control would have been chromatin states in somatic follicle cells of the ovary, since we know germ cell genes such as *ovo* and *otu* are not expressed and presumably the chromatin states in somatic cell types would be different than germ cells. However, chromatin states for somatic follicle cells were not published in these two papers and we are not aware of any other existing datasets to compare too. Essentially, we need to determine the changes in chromatin states with and without OVO, which we are currently working on.

We did further analyze chromatin states and differential OVO binding in respect to gene elements, and found that OVO binding, regardless of the relationship to the gene element, is always open (gsc and 32c ATAC). OVO binding over the gene body shows the same enrichment for open chromatin and transcriptionally active histone marks. We compared the profiles of these chromatin marks and the promoters of OVO bound and not bound genes and consistent with the suggestion that promoters are generally open, we found that this was the case. However, there is an enrichment for open chromatin and transcriptionally active histone marks for OVO bound genes compared to non-OVO bound genes. This could be a consequence of OVO binding or indirect consequence of a downstream OVO target. Regardless, as has been suggested, future experiments directly measuring chromatin status and OVO needs to be performed. The below excerpts have been added to the text to supplement the comments provided above.

Lines 328-343

“The association of OVO binding with active histone marks and open chromatin was striking, but open chromatin is likely a general phenomenon of promoters (Haines and Eisen, 2008). Indeed, when measuring the read density for GSC and 32C ATAC-seq for OVO bound and OVO non-bound promoters, there is an enrichment for open chromatin at the TSS regardless of OVO binding. However, we did notice an increase in enrichment for OVO bound promoters compared to OVO non-bound promoters (Figure S1G), possibly suggesting that OVO bound promoters are more open or have an increase in accessibility when compared to non-OVO bound promoters. This same relationship held true for the transcriptionally active histone mark H3K27ac in GSCs (Figure S1H). Since only 45% of OVO ChIP peaks overlapped TSSs, we plotted the read density of the above chromatin marks over OVO ChIP peak maximums for OVO bound over the TSS, gene body, or intergenic regions (Figure S2A-D). We found that OVO bound regions that were not overlapping the TSS still showed the same propensity for enrichment of open chromatin and active histone marks. Intergenic regions were especially enriched for open chromatin measured through ATAC-seq. Altogether suggesting that OVO binding genome-wide is tightly associated with open chromatin regardless of germ cell stage, and active transcription in GSCs. In other words, chromatin state data suggests OVO is acting positively on its target genes and raises the possibility that OVO-binding and open chromatin are related.”

For clarity, it would help the reader if the authors mentioned the male-specific TATA-associated factors as a rationale for testing the role of OVO binding in core promoter function. This is currently mentioned in the Discussion on lines 575-577, but would help in understanding the motivation behind the detailed analysis of the promoter binding of OVO in the Results and make the negative result more clearly impactful.

We have introduced the male specific tata factors as suggested and have condensed the two intro paragraphs in this section into one, as shown below.

Lines 347-363

“Our data thus far clearly indicates that OVO binding occurs at or very near the core promoter, a region recognized by an enormous collection of factors that associate with RNA polymerase to initiate transcription (Aoyagi and Wassarman 2000; Vo Ngoc, Kassavetis, and Kadonaga 2019). The highly organized polymerase complex has sequence-specific DNA recognition sites with incredibly precise spacing between them, with an overall DNA footprint of a little less than 100bp (Rice, Chamberlin, and Kane 1993; FitzGerald et al. 2006; Ohler et al. 2002). There are upstream binding sites such as TATA, sites at transcription start, such as the initiator (INR), and downstream promoter elements (DPE) (Vo Ngoc, Kassavetis, and Kadonaga 2019). The combinations of these DNA motifs is not random in mammals and Drosophila (FitzGerald et al. 2006), and distinct combinations of different motifs at the TSS of genes expressed in *Drosophila* are conserved over tens of millions of years of evolution (Chen et al. 2014). The male germline expresses a number of TATA-associated factors that have been implicated in male-specific promoter usage for gene expression (M. Hiller et al. 2004; M. A. Hiller et al. 2001; Lu et al. 2020; V. C. Li et al. 2009). It is possible that OVO is a female germline specific TATA-associated factor, and if so, OVO binding sites at core promoters should share precise spacing with other core promoter elements, suggesting it is likely part of the complex. If not, then OVO is more likely to facilitate binding of the basal transcriptional machinery. Because of the extended footprint of engaged RNA polymerase, OVO and the basal machinery would not be likely to occupy the same region at the same time.”

The description of the system used for the RNA-seq would benefit from additional clarity. It is not clear as written why it is "Lucky" that there is an mRNA isoform with extended exon 2 required for egg chamber development beyond stage 5. How does this requirement compare to the global requirement for OVO, which seems to be required for germ cell development even before stage 5? Understanding this system is essential for interpreting the RNA-seq results. Indeed, the authors have a separate manuscript (currently on bioRxiv) that explains the details of this system. As such, the current description requires that the reader refer to this additional pre-print. Could the authors include a diagram to better illustrate this system? Furthermore, since this RNA-seq is being performed on tissue that includes nurse cells, follicle cells, and germ cells from multiple stages of development, it is important for the authors to clearly state in which cell types OVO is expressed and likely functional. (While this is well beyond this manuscript, this analysis is the type that might benefit from the use of single-cell sequencing as a means to deconvolute the phenotypic effects of OVO loss.)

We have rewritten the text to better describe the system for RNA-seq. We have also included a figure (Figure S1A) showing the alleles used that should help provide clarity for the readers. We agree that moving forward single cell experiments will be critical to have a better understanding of the transcriptional changes and chromatin dynamics with and without OVO. We have included the below changes to the text.

Lines 409-423

“Previous work from our lab has identified a transheterozygous *ovo* allelic combination (*ovoovo-GAL4*/*ovoΔBP*) that greatly reduces OVO activity resulting in sterility, however, female germ cells are able to survive up until at least stage 5 of oogenesis (Benner et al. 2023). *ovoovo-GAL4* is a CRISPR/Cas9 derived T2A-GAL43xSTOP insertion upstream of the splice junction of exon 3 in the *ovo-RA* transcript (Figure S1A).

Importantly, this insertion in the extended exon 3 would disrupt roughly 90% of the *ovo-B* transcripts. However, since about 10% of *ovo-B* transcripts utilize an upstream splice junction in exon 3, these transcripts would not be disrupted with the T2A-GAL4-3xSTOP insertion and thus allow for enough OVO activity for germ cell survival (Benner et al. 2023). Since *ovoovo-GAL4* expresses GAL4 in place of full length OVO due to the T2A sequences, we can drive expression of a rescuing OVO-B construct downstream of *UASp* to generate OVO+ female germ cells, which in fact does rescue the arrested germ cell phenotype of *ovoovo-GAL4*/*ovoΔBP* ovaries. Therefore, in order to determine genes that are transcriptionally responsive to OVO, we compared the gene expression profiles in sets of ovaries that had the *ovo* hypomorphic phenotype with a negative control rescue construct (*ovoovo-GAL4*/*ovoΔBP*; *UASp-GFP*)(Figure 4A) versus those that drive expression of the rescue construct expressing OVO-B (*ovoovo-GAL4*/*ovoΔBP*; *UASp-3xFHAOVO-B*)(Figure 4B).”

Lines 427-432

“The adult female ovary contains somatic cells, germline stem cells, and germline derived nurse cells that would be profiled in a bulk ovary tissue RNA-seq experiment. Although OVO is only required and expressed in germline derived cell types, we chose to dissect one day old post-eclosion *ovoovoGAL4*/*ovoΔBP*; *UASp-3xFHA-OVO-B* female ovaries to enrich for early stages of oogenesis and collected only ovarioles containing the germarium through previtellogenic egg chambers.”

On lines 526-532, it is unclear why the genes fs(1)N, fs(1)M3, and closca are particularly sensitive to the ovoD3 allele. What is this allele trans heterozygous with in the assay that allows development through egg laying? Why might these genes be unique in their sensitivity?

These genes are not particularly sensitive, the transheterozygous hypomorphic *ovo* ovaries are weak enough to reveal the role of OVO for these genes. We rewrote this paragraph to try and provide more clarity to the relationship between OVO+ binding at these vitelline membrane genes and the phenotype of OVOD3 expressing females.

Lines 562-577

“We also found that the genes *fs(1)N*, *fs(1)M3*, and *closca*, were all bound by OVO and responded transcriptionally to the presence of ectopic rescue OVO. These genes are significant because they constitute a set of genes that are expressed in the germline and the encoded proteins are eventually incorporated into the vitelline membrane providing the structural integrity and impermeability of the egg (Mineo, Furriols, and Casanova 2017; Ventura et al. 2010). Loss-of-function of these three genes results in flaccid eggs that are permeable to dye and fail to develop. The loss-of-function phenotype of *fs(1)N*, *fs(1)M3*, and *closca* closely resembles the dominant antimorph *ovoD3* phenotype. The *ovoD3* allele is the weakest of the original dominant-negative *ovo* alleles and produces defective eggs allowing us to explore the role of OVO in late stages (Busson et al. 1983; Komitopoulou et al. 1983). *ovoD3*/ovo+ transheterozygous females express a repressive form of OVO that results in dominant sterility, and importantly, these females lay flaccid eggs with compromised vitelline membranes that are permeable to the dye neutral red (Oliver, Pauli, and Mahowald 1990). Since OVO+ is bound at the TSS of *fs(1)N*, *fs(1)M3*, and *closca*, and these three genes respond transcriptionally to OVO+, then it is plausible that the repressive OVOD3 is negatively regulating these three genes that are required for vitelline membrane formation. This is evidence that OVO is not only involved in regulating the expression of numerous essential maternal pathways for embryonic development, but it is also essential for regulating genes that are required for egg integrity and maturation.”

The Discussion of OVO as a pioneer factor is highly speculative and based only on correlative data. In fact, the expression data in the embryonic germline is not included in this manuscript, but rather in a separate bioRxiv preprint. This makes it challenging to understand, why this is extensively discussed here. However, there are experiments that could begin to test this proposal. OVO could be expressed in an exogenous tissue and test whether it promotes accessibility. Also, mutations could be made (using gene editing) to identify previously known OVO binding sites in the otu and/or other promoters and these could be assayed for accessibility. By selecting promoters of genes that are not essential for germline development, the authors could directly test the role of OVO in promoting chromatin accessibility. Alternatively, are there reasons that the system used for RNA-seq couldn't be similarly used for ATACseq? It is imperfect but could provide insights into chromatin accessibility in the absence of OVO.

We have largely removed the speculation on pioneering activity, reference to embryonic germline OVO dynamics included in the previous work, and Figure 7. These are excellent suggestions for experiments and ones we are currently pursuing. Below is the modified discussion.

Lines 645-663

“The requirement for OVO at the TSS of target genes has been well characterized at its own locus as well as its downstream target *otu*. Our OVO ChIP and expression data confirm findings from previous work that OVO is binding to these target promoters, and in the case of *otu*, strongly responds transcriptionally to the presence of OVO. Although we did not test the requirement for OVO DNA binding motifs at other OVO bound genes in this work, this has been extensively explored before, showing that removal of OVO

DNA binding sites overlapping the TSS results in a strong decrease in reporter expression (Lü et al. 1998; Bielinska et al. 2005; Lü and Oliver 2001). Removal of more distal upstream OVO DNA binding sites also reduces reporter expression to a lesser degree. However, for most cases tested, removal of OVO DNA binding sites while leaving the rest of the enhancer regions intact, never totally abolished reporter expression. These dynamics are highly similar to work that has been completed on the pioneer factor *zelda* (*zld*). Adding *zld* DNA binding motifs to a stochastically expressed transcriptional reporter increases the activity and response of the reporter (Dufourt et al. 2018). Distally located *zld* DNA binding motifs influenced reporter expression to a lesser degree than proximal sites. A single *zld* DNA binding site adjacent to the TSS produced the strongest reporter activity. Importantly, just like the activity of OVO transgenic reporters, there is not an absolute requirement for *zld* DNA binding to activate reporter expression, however, the addition of TSS adjacent *zld* DNA binding motifs does strongly influence reporter response. We know that *zld* achieves this reporter response through its pioneering activity (Xu et al. 2014; Harrison et al. 2011), whether OVO achieves this similar effect on gene expression through a shared mechanism, or in cooperation with other transcription factors needs to be further explored.”

The authors suggest that OVO binding is essential for transcriptional activation, but that this may be indirect and that expression of other transcription factors might be necessary for activating gene expression. Did the motif analysis of the OVO-bound regions suggest additional transcription factors that might provide this function?

We did find other motifs significantly enriched in OVO ChIP peaks. We performed XSTREME analysis on the same set of OVO ChIP peaks which allowed us to determine if any of these motifs were significant matches to DNA binding motifs of known transcription factors. Notably, the DNA binding motifs of GAF and CLAMP were enriched in OVO ChIP peaks. GAF is required in germline clones and the potentially for co-regulation of genes is possible. Other enriched motifs did not match any known binding motifs of other transcription factors but we reported some of the most significantly enriched motifs that were alongside of OVO in Figure S1C-F. The below text outlines changes made to the text incorporating these findings.

Lines 170-182

“Along with the OVO DNA binding motif, other motifs were also significantly enriched in OVO ChIP peaks. The motif 5’-GWGMGAGMGAGABRG-3’ (Figure S1C) was found in 18% of OVO ChIP peaks and is a significant match to the DNA binding motifs of the transcription factors GAF (*Trl*) (Omelina et al. 2011) and CLAMP (Soruco et al. 2013). *Trl* germline clones are not viable, indicating that GAF activity is required in the germline during oogenesis (Chen et al. 2009). The possibility that OVO binds with and regulates genes alongside of GAF given the enrichment of both transcription factors DNA binding motifs is intriguing. Other significantly enriched motifs 5’-ACACACACACACACA-3’ (29% of peaks, Figure S1D), 5’-RCAACAACAACAACA-3’ (26% of peaks, Figure S1E), and 5’-GAAGAAGAAGAAGAR-3’ (17% of peaks, Figure S1F) were present in OVO ChIP peaks, however, these motifs did not significantly match known

DNA binding motifs of other transcription factors. Determining the factors that bind to these sequences

will certainly help elucidate our understanding of transcriptional control with relationship to OVO in the female germline.”

The figures would benefit from a bit more detail in the legends (see comments below).Minor comments:In multiple places throughout the document, the citations are inadvertently italicized (see lines 57-59, 91, and 327 as examples.)

We have changed this in these locations and other instances in the text.

On line 76, when discussing OVO as a transcription factor this is referencing the protein and not the gene. Thus, should be written OVO and not ovo.

We have made the correction *ovo* to OVO.

On line 349, "core" promoters is likely what is meant rather than "care" promoters.

We have corrected ‘care’ to ‘core’ in the text.

On line 404, the authors state that they wanted to use a "less conservative log2 fold change" but it is not clear what they are comparing to. This is important to understand the motivation.

We are talking about the gene expression comparison between the ectopic *ovo* rescue and *ovo* hypomorphic ovaries. “less conservative” was an unfortunate phrasing. We have rewritten the text to state this directly to the reader.

Lines 435-444

“We then performed RNA-seq in quadruplicate and measured the changes in gene expression between ectopic rescue OVO and hypomorphic OVO ovaries. We used a significance level of p-adj < 0.05 and a log2 fold change cutoff of >|0.5| to call differential expression between these two sets of ovaries. We utilized these log2 fold change cutoffs for two reasons. Our control ovary genotype (*ovoovo-GAL4*/*ovoΔBP*; *UASp-GFP*) has hypomorphic OVO activity, hence germ cells can survive but are arrested. With the addition of ectopic rescue OVO in *ovoovo-GAL4*/*ovoΔBP*; *UASp-3xFHA-OVO-B* ovaries, we predicted that genes that were directly regulated by OVO would transcriptionally respond, however, we were unsure as to what degree the response would be in comparison to hypomorphic OVO. We reasoned that if the changes were not significant between genotypes, then minor changes in gene expression would not matter.”

On line 615, it is unclear what is meant by "showing expression with only 10s of bp of sequence in reporters."

This is in reference to some of the previously studied ovo reporter deletion lines, however, we have decided to remove the below text in the revised discussion.

“, despite being remarkably compact. The OVO-dependent *ovo* core promoter is very compact; showing expression with only 10s of bp of sequence in reporters.”

It would be useful to cite and discuss Dufourt et al. Nature Communications 2018 (PMID30518940) regarding the role of Zelda in potentiating transcriptional activation when mentioned on line 624.

We have added this and the relationship to previous similar work on OVO in the discussion.

Lines 645-663

“The requirement for OVO at the TSS of target genes has been well characterized at its own locus as well as its downstream target *otu*. Our OVO ChIP and expression data confirm findings from previous work that OVO is binding to these target promoters, and in the case of *otu*, strongly responds transcriptionally to the presence of OVO. Although we did not test the requirement for OVO DNA binding motifs at other OVO bound genes in this work, this has been extensively explored before, showing that removal of OVO

DNA binding sites overlapping the TSS results in a strong decrease in reporter expression (Lü et al. 1998; Bielinska et al. 2005; Lü and Oliver 2001). Removal of more distal upstream OVO DNA binding sites also reduces reporter expression to a lesser degree. However, for most cases tested, removal of OVO DNA binding sites while leaving the rest of the enhancer regions intact, never totally abolished reporter expression. These dynamics are highly similar to work that has been completed on the pioneer factor *zelda* (*zld*). Adding *zld* DNA binding motifs to a stochastically expressed transcriptional reporter increases the activity and response of the reporter (Dufourt et al. 2018). Distally located *zld* DNA binding motifs influenced reporter expression to a lesser degree than proximal sites. A single *zld* DNA binding site adjacent to the TSS produced the strongest reporter activity. Importantly, just like the activity of OVO transgenic reporters, there is not an absolute requirement for *zld* DNA binding to activate reporter expression, however, the addition of TSS adjacent *zld* DNA binding motifs does strongly influence reporter response. We know that *zld* achieves this reporter response through its pioneering activity (Xu et al. 2014; Harrison et al. 2011), whether OVO achieves this similar effect on gene expression through a shared mechanism, or in cooperation with other transcription factors needs to be further explored.”

On line 1006 (Figure 1 legend), it is unclear what is meant by "The percentage of OVO ChIP peaks each motif was found". Is a word missing?

This was unclear, we have revised the sentence below.

Lines 1035-1036

“The percentage of OVO ChIP peaks containing each motif and their corresponding p-value are indicated to the right.”

In the Figure 1 legend, please include citations for the Garfinkel motif and Oliver motif.

Included, as below.

Lines 1036-1039

“H OVO ChIP minus input control ChIP-seq read coverage density centered on the location of the four de novo OVO DNA binding motifs and previously defined in vitro OVO DNA binding motifs (Lü et al. 1998, Bielinska et al. 2005, Lee and Garfinkel 2000).”

In Figure 2 legend, it is unclear if B is all instances of a given motif or the DNA motifs that are bound by ChIP. Please clarify.

We meant only the OVO DNA binding motifs that were within significant OVO ChIP peaks. We have revised the legend below.

Lines 1049-1052

“A, B OVO ChIP minus input control, GSC and 32c ATAC-seq, GSC H3K27ac, H3K4me3, H3K27me3, H3K9me3, 8c NC H3K9me3, 32c NC H3K27ac, and H3K27me3 ChIP-seq read coverage density centered on each OVO peak maximum or OVO DNA binding motif located within a significant OVO ChIP peak.”

The Figure legend for 2D could use more explanation. What do the lines and circles indicate?

These lines and circles indicate the amount of overlapping peaks measured between the two datasets with solid circles. We have included a better description of what these indicate in the figure legend.

Lines 1054-1058

“D Total number of significant peaks (left) and the total number of overlapping peaks (top) between OVO

ChIP and GSC and 32c ATAC-seq, GSC H3K27ac, H3K4me3, H3K27me3, H3K9me3, 8c NC H3K9me3, 32c NC H3K27ac, and H3K27me3 ChIP-seq. Lines connecting solid dots indicates the amount of overlapping peaks between those two corresponding datasets.”

In Figure 4C, bring the 564 blue dots forward so they are not masked by the yellow dots.

We have brought the colored dots forward in both figure 4C and 4D.

In Figure 4E, what is the order of the heatmaps?

The order is genes with the highest to lowest OVO read density enrichment. We have included this in the figure 4 legend.

Lines 1086-1087

“The order of the heatmap is genes with the highest to lowest amount of OVO ChIP read density.”

In Figure 5, the order of the tracks is not immediately obvious. It appears to be those chromatin features most associated with OVO ChIP and those less correlated. Additional clarity could be provided by showing these tracks (and in Supplemental Figure S2) in different colors with a reference to the figure legend about what the colors might indicate.

We have changed the colors and order of the tracks to be more similar and consistent in both figures.

Lines 1090-1093

“*ovo* gene level read coverage tracks for OVO ChIP minus input (black), GSC and 32c ATAC-seq (light blue), GSC and 32C H3K27ac (green), H3K4me3 (dark blue), GSC and 32c H3K27me3 (orange), and GSC and 8c H3K9me3 (pink) ChIP-seq, and *ovoΔBP*/*ovoovo-GAL4*; *UASp-3xFHA-OVO-B* minus *ovoΔBP*/*ovoovo-GAL4*; *UASp-GFP* RNA-seq (red).”

In Figure S1 legend, what is the reference to the da-GAL4 X UAS transgene in the title?

This was an error on our part and we have removed it.

**Reviewer #2 (Recommendations For The Authors):**
Overall, the manuscript would benefit from revisions of the writing style. At times it is difficult to distinguish between hypothesis and results. The use of colloquial phrases/prose was distracting while reading, which the authors may consider revising. Some sentences were confusing or extraneous, and the authors may consider revising those. Occasionally sentences within the results sections seem more appropriate for the materials and methods.(1) The manuscript is generally clear; however, it is at times difficult to distinguish between hypothesis and results. The use of colloquial phrases/prose was distracting while reading, which the authors may consider revising. Examples include:a) Lines 48-49 "While thematic elements of this complex orchestration have been well studied, coordinate regulation of the symphony has not."

We have edited this sentence below.

Lines 48-50

“While the complex interactions between maternally supplied mRNAs and proteins have been well studied, transcriptional regulation driving the expression of these pathways are less well understood.“

b) Lines 232-233 "In other words, where exactly does transcription start at these genes."

We have removed this sentence.

c) Line 385, the word "sham" could be changed to "negative control" or "GFP control"

We have rewritten this sentence below.

Lines 419-423

“Therefore, in order to determine genes that are transcriptionally responsive to OVO, we compared the gene expression profiles in sets of ovaries that had the *ovo* hypomorphic phenotype with a negative control rescue construct (*ovoovo-GAL4*/*ovoΔBP*; *UASp-GFP*)(Figure 4A) versus those that drive expression of the rescue construct expressing OVO-B (*ovoovo-GAL4*/*ovoΔBP*; *UASp-3xFHA-OVO-B*)(Figure 4B)”

d) Line 490 "For the big picture"

We have removed this and revised with the below sentence.

Lines 530-531

“To do this, we performed Gene Ontology enrichment analysis with gProfiler software (Raudvere et al. 2019).”

(2) Some sentences were confusing or extraneous, and the authors may consider revising them. Examples include:a) Lines 195-196 "Therefore, we plotted the significant ChIP (minus input) read density peaks centered on the location of the motif itself."

We have removed the word ‘peaks’ and ‘itself’, as below.

Lines 200-201

“Therefore, we plotted the significant ChIP (minus input) read density centered on the location of the motif.”

b) Lines 201-203 "... over the location of the motifs, strongly reinforces the idea that our dataset contains regions centered on sequence-specifically bound OVO transcription factor in the ovary."

We have edited this sentence to clarify below.

Lines 204-208

“While it is possible that OVO comes into contact with regions of DNA in three-dimensional nuclear space non-specifically, the presence of OVO motifs within a large percentage of significant ChIP peaks in vivo and enrichment of OVO ChIP read density at the location of the motifs, strongly reinforces the idea that our OVO ChIP dataset contains regions centered on sequences specifically bound by OVO in the ovary.”

c) Lines 326-328 "The combinations of these elements...tens of millions of years of evolution."

We have revised this sentence below.

Lines 354-357

“The combinations of these DNA motifs is not random in mammals and Drosophila (FitzGerald et al. 2006), and distinct combinations of different motifs at the TSS of genes expressed in *Drosophila* are conserved over tens of millions of years of evolution (Chen et al. 2014).”

d) Lines 444-446 "To address this directly, we tested the idea that genes with... and thus downstream of OVO."

We have removed this sentence in its entirety.

e) Line 579-580 "Where OVO binding in close proximity, in any ...activates transcription"

We have removed this sentence in its entirety.

(3) Occasionally sentences within the results sections seem more appropriate for the materials and methods. For example, lines 213-218.(4) At the end of line 375, do the authors mean "only" instead of "also"?

We have modified this sentence below.

Lines 411-414

“*ovoovo-GAL4* is a CRISPR/Cas9 derived T2A-GAL4-3xSTOP insertion upstream of the splice junction of exon 3 in the *ovo-RA* transcript (Figure S1A). Importantly, this insertion in the extended exon 3 would disrupt roughly 90% of the *ovo-B* transcripts. However, since about 10% of *ovo-B* transcripts utilize an upstream splice junction in exon 3, these transcripts would not be disrupted with the T2A-GAL4-3xSTOP insertion and thus allow for enough OVO activity for germ cell survival (Benner et al. 2023).”

(5) In line 392 the authors say that they dissected ovaries "one day post-eclosion" but the methods section says that ovaries were 3-5 days old. Please clarify.

We meant one day old for the RNAseq experiments. We have changed this in the text.

Lines 679-681

“Twenty, one day old post-eclosion *ovoΔBP*/*ovoovo-GAL4*; *UASp-GFP* and *ovoΔBP*/*ovoovo-GAL4*; *UASp-3xFHAOVO-B* ovaries were dissected and germariums through previtellogenic egg chambers were removed with microdissection scissors and placed in ice cold PBS making up one biological replicate.”

(6) In line 668 the authors mention CRISPR/Cas9 in the methods, but no such experiment was described.

We have removed this from the Methods header.